# An Erg-driven transcriptional program controls B cell lymphopoiesis

Ashley P. Ng [1,2✉], Hannah D. Coughlan[2,3], Soroor Hediyeh-zadeh [3], Kira Behrens [1],
Timothy M. Johanson[2,4], Michael Sze Yuan Low[2,4,5], Charles C. Bell [6,7], Omer Gilan[6,7], Yih-Chih Chan [6,7],
Andrew J. Kueh[1,2], Thomas Boudier [2,8], Rebecca Feltham[9], Anna Gabrielyan[9], Ladina DiRago[1],
Craig D. Hyland [1], Helen Ierino[1], Sandra Mifsud[1], Elizabeth Viney[1], Tracy Willson[1], Mark A. Dawson [6,7,10],
Rhys S. Allan [2,4], Marco J. Herold[1,2], Kelly Rogers[2,8], David M. Tarlinton [11], Gordon K. Smyth [2,3],
Melissa J. Davis [2,3], Stephen L. Nutt [2,4] & Warren S. Alexander[1,2]

B lymphoid development is initiated by the differentiation of hematopoietic stem cells into lineage committed progenitors, ultimately generating mature B cells. This highly regulated process generates clonal immunological diversity via recombination of immunoglobulin V, D and J gene segments. While several transcription factors that control B cell development and V(D)J recombination have been defined, how these processes are initiated and coordinated into a precise regulatory network remains poorly understood. Here, we show that the transcription factor ETS Related Gene (*Erg*) is essential for early B lymphoid differentiation. Erg initiates a transcriptional network involving the B cell lineage defining genes, *Ebf1* and *Pax5*, which directly promotes expression of key genes involved in V(D)J recombination and formation of the B cell receptor. Complementation of Erg deficiency with a productively rearranged immunoglobulin gene rescued B lineage development, demonstrating that Erg is an essential and stage-specific regulator of the gene regulatory network controlling B lymphopoiesis.

[1] Blood Cells and Blood Cancer Division, The Walter and Eliza Hall Institute of Medical Research, Parkville, VIC 3052, Australia. [2] Department of Medical Biology, The University of Melbourne, Parkville, VIC 3010, Australia. [3] Bioinformatics Division, The Walter and Eliza Hall Institute of Medical Research, Parkville, VIC 3052, Australia. [4] Immunology Division, The Walter and Eliza Hall Institute of Medical Research, Parkville, VIC 3052, Australia. [5] Monash Haematology, Monash Hospital, Clayton, VIC 3004, Australia. [6] Peter MacCallum Cancer Centre, Parkville, VIC 3000, Australia. [7] Sir Peter MacCallum Department of Oncology, The University of Melbourne, Parkville, VIC 3010, Australia. [8] Advanced Technology and Biology Division, The Walter and Eliza Hall Institute of Medical Research, Parkville, VIC 3052, Australia. [9] Inflammation Division, The Walter and Eliza Hall Institute of Medical Research, Parkville, VIC 3010, Australia. [10] Centre for Cancer Research, The University of Melbourne, Parkville, VIC 3010, Australia. [11] Department of Immunology and Pathology, Monash University, Melbourne, VIC 3004, Australia. ✉email: ang@wehi.edu.au

Transcription factors are critical for controlling the expression of genes that regulate B-cell development. The importance of specific B-lymphoid transcription factors is highlighted by the phenotype of gene knockout models. Failure of B-cell lineage specification from multi-potential progenitors occurs with deletion of *Ikzf1*[1] and *Spi1 (Pu.1)*[2], while deletion of *Tcf3 (E2A)*[3] and *Foxo1*[4] results in failure of B-cell development from common lymphoid progenitors (CLPs). Developmental arrest later in B lymphopoiesis is observed with deletion of *Ebf1* and *Pax5* at the pre–proB and proB stages, respectively[5,6]. This sequential pattern of developmental arrest associated with the loss of gene function, along with ectopic gene complementation studies[2], gene expression profiling[7] and analysis of transcription factor binding to target genes, support models in which transcription factors are organised into hierarchical gene regulatory networks that specify B-lymphoid lineage fate, commitment and function[8].

Two transcription factors that have multiple roles during B-cell development are Ebf1, a member of the COE family, and Pax5, a member of the PAX family. While Ebf1 and Pax5 have been shown to bind to gene regulatory elements of a common set of target genes in a co-dependent manner during later stages of B lineage commitment[9], both manifest distinct roles during different developmental stages. Ebf1 has been proposed to form a transcriptional network with E2A and Foxo1 in CLPs that appears important in early B-lymphoid fate determination[10], while during later stages of B lymphopoiesis, Ebf1 acts as a pioneer transcription factor that regulates chromatin accessibility at a subset of genes co-bound by Pax5[11] as well as at the *Pax5* promoter itself[12]. Pax5 in contrast, regulates B-cell genomic organisation[13] including the *Immunoglobulin heavy chain (Igh)* locus during V(D)J recombination, co-operating with factors such as CTCF[14], as well as transactivating[15] and facilitating the activity of the recombinase activating gene (Rag) complex[16].

It is unclear, however, how these various functions of Ebf1 and Pax5 are co-ordinated during different stages of B-lymphoid development. In particular, it would be important to ensure co-ordinated *Ebf1* and *Pax5* co-expression before the pre-BCR checkpoint, such that *Ebf1* and *Pax5* co-regulated target genes required for V(D)J recombination and pre-B-cell receptor complex formation are optimally expressed[9].

Here we show that the ETS-related gene (*Erg*), a member of the ETS family of transcription factors, plays this vital role in B lymphopoiesis. Deletion of *Erg* from early lymphoid progenitors resulted in developmental arrest at the early pre–proB-cell stage and loss of $V_H$-to-$DJ_H$ recombination. Gene expression profiling, DNA-binding analysis and complementation studies demonstrated Erg to be a transcriptional regulator that lies at the apex of an Erg-dependent Ebf1 and Pax5 gene regulatory network commencing in pre–proB cells. This co-dependent transcriptional network directly controls expression of the *Rag1/Rag2* recombinase activating genes and the *Lig4* and *Xrcc6* DNA repair genes required for V(D)J recombination, as well as expression of components of the pre-BCR complex such as *CD19, Igll1, Vpreb1* and *Vpreb2*. Taken together, we define an essential Erg-mediated transcription factor network required for regulation of *Ebf1* and *Pax5* expression that is exquisitely stage specific during early B-lymphoid development.

## Results

### *Erg* is required for B-cell development.
To build on prior work defining the role of the transcription factor *Erg* in regulation of hematopoietic stem cells (HSCs)[17] and megakaryocyte-erythroid specification[18], we sought to identify whether *Erg* played roles in other hematopoietic lineages. *Erg* expression in adult hematopoiesis was first examined by generating mice carrying the $Erg^{tm1a(KOMP)wtsi}$ knock-in first reporter allele ($Erg^{KI}$) (Fig. 1a). Consistent with the known role for *Erg* in hematopoiesis[17–21], significant *LacZ* expression driven by the endogenous *Erg* promoter was observed in HSCs and multi-potential progenitor cells, as well as in granulocyte-macrophage and megakaryocyte-erythroid progenitor populations, with declining activity accompanying erythroid maturation (Fig. 1b with definitions of cells examined provided in Supplementary Table 1 and representative flow cytometry plots in Supplementary Fig. 1). In other lineages, transcription from the *Erg* locus was evident in CLP, all lymphoid and B-cell-biased lymphoid progenitor cells, as well as in B lineage committed pre–proB, proB and preB cells and double-negative thymic T-lymphoid cell subsets, with a reduction in transcription with later B- and T-cell maturation (Fig. 1b, c). We confirmed these findings with RNA-sequencing (RNA-seq) analysis that showed significant *Erg* RNA in pre–proB, proB and preB cells (Fig. 1d). This detailed characterisation of *Erg* expression raised the possibility that *Erg* plays a stage-specific function at early developmental stages of the lymphoid lineages.

To determine whether *Erg* had a role in lymphoid development, mice carrying floxed *Erg* alleles ($Erg^{fl/fl}$, Fig. 1a) were interbred with *Rag1Cre* transgenic mice that efficiently delete floxed alleles in CLPs and T- and B-committed progenitor cells[22], but have normal lymphoid development (Supplementary Fig. 2a). The resulting $Rag1Cre^{T/+};Erg^{\Delta/\Delta}$ mice specifically lack *Erg* throughout lymphopoiesis (Fig. 1e, Supplementary Fig. 2b). While numbers of red blood cells, platelets and other white cells were normal, $Rag1Cre^{T/+};Erg^{\Delta/\Delta}$ mice displayed a deficit in circulating lymphocytes (Supplementary Table 2). This was due to a specific absence of B cells; the numbers of circulating T cells and thymic progenitors were not decreased (Fig. 1f, Supplementary Fig. 2c).

B cells are produced from bone marrow progenitor cells that progress through regulated developmental stages. B-lymphoid development was markedly compromised in $Rag1Cre^{T/+};Erg^{\Delta/\Delta}$ mice, with proB, preB, immature B and mature recirculating B cells (Hardy fractions C-F, defined in Supplementary Table 1) markedly reduced in number or virtually absent (Fig. 1f). A B-lymphoid developmental block was clearly evident at the pre–proB (Hardy fraction A-to-B) stage, with excess numbers of these cells present in the bone marrow.

### *Erg* deficiency perturbs $V_H$-to-$DJ_H$ recombination.
To further characterise the developmental B lineage block in $Rag1Cre^{T/+}$; $Erg^{\Delta/\Delta}$ mice, $B220^+$ bone marrow cells were examined for *Igh* somatic recombination. Unlike cells from control $Erg^{fl/fl}$ mice, $B220^+$ cells from $Rag1Cre^{T/+};Erg^{\Delta/\Delta}$ mice had not undergone significant $V_H$-to-$DJ_H$ immunoglobulin heavy chain gene rearrangement, although $D_H$-to-$J_H$ recombination was relatively preserved (Fig. 2a).

We next investigated the abnormalities underlying *Igh* recombination in greater detail. We first undertook fluorescence in situ hybridisation (FISH) at the *Igh* locus to measure the intra-chromosomal distance between distal $V_HJ558$ and proximal $V_H7183$ $V_H$ family genes, as cell stage-specific contraction of the *Igh* locus is essential for efficient V(D)J recombination[23]. This revealed that pre–proB cells from $Rag1Cre^{T/+};Erg^{\Delta/\Delta}$ mice had reduced locus contraction compared with $Erg^{fl/fl}$ controls (Fig. 2b). To assess whether other structural perturbations across the *Igh* locus were also present, high throughput chromatin conformation capture (in situ Hi-C) was performed. We performed a differential analysis of the data and revealed a reduction of long-range interactions across the *Igh* locus in $Rag1Cre^{T/+};Erg^{\Delta/\Delta}$ pre–proB cells when

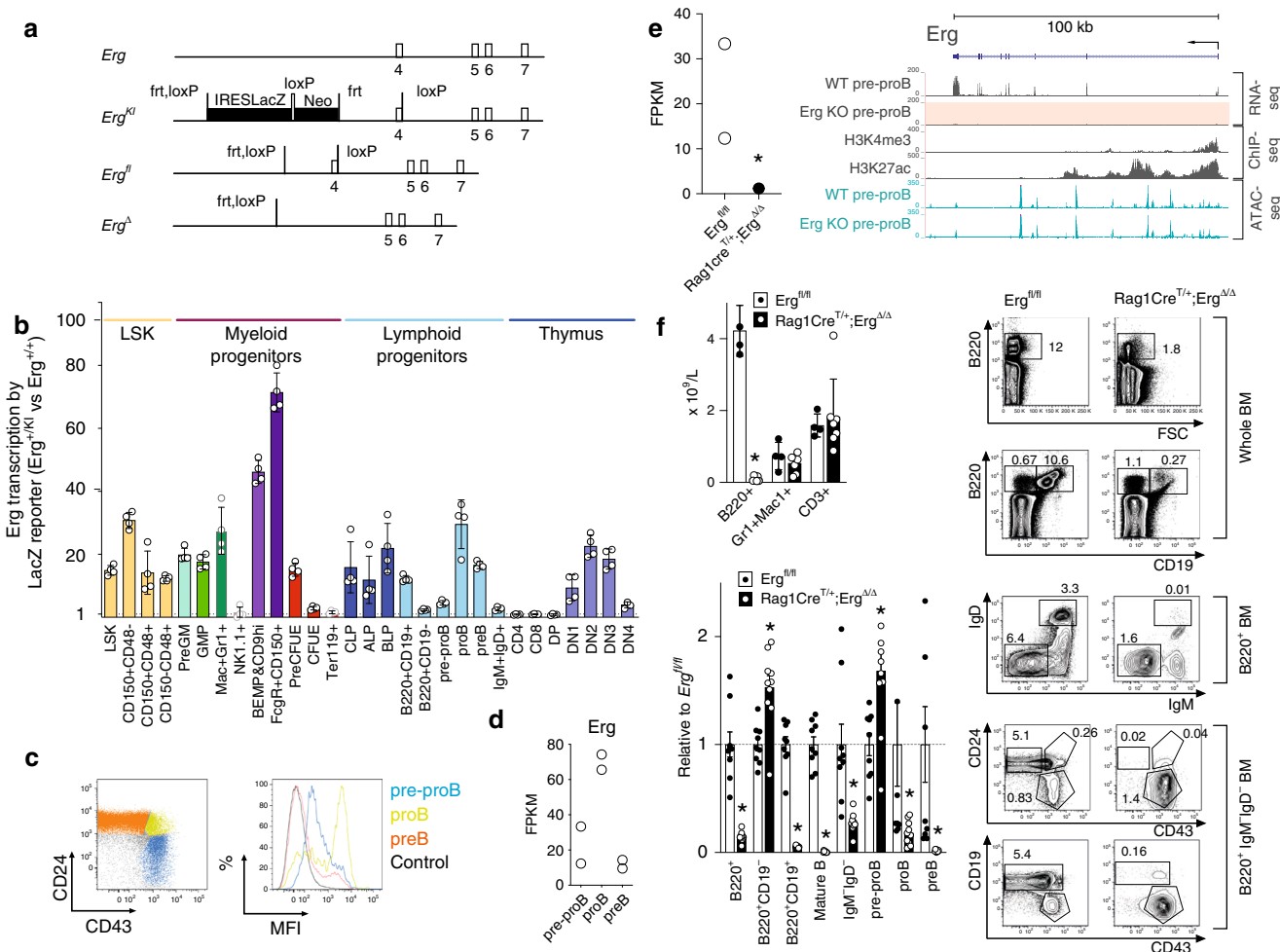

**Fig. 1 Expression and targeted disruption of *Erg* in lymphopoiesis. a** Wild-type (*Erg*), *Erg*tm1a(KOMP)wtsi *lacZ* reporter (*Erg*KI), conditional (*Erg*fl), and Cre recombinase-deleted (*Erg*Δ) alleles with exons, Cre (loxP) and Flp (frt) recombinase recognition sites. IRES, internal ribosome entry site; Neo, neomycin-resistance cassette. **b** *Erg* transcriptional activity by *lacZ* expression in *Erg*KI bone marrow (BM) and thymus cell populations (see "Methods" and Supplementary Fig. 1, Supplementary Table 1). Mean fluorescent intensity (MFI) ratio mean ± S.D of *Erg*KI (n = 4) to C57BL/6 (n = 4) biologically independent samples. $P_{adj} < 0.028$ by the Student's two-tailed unpaired *t* test corrected using the Holm's modification for multiple testing for each population except BM Ter119+ and NK1.1+, and thymic DP, CD4+CD8− and CD8+CD4− populations ($P_{adj} > 0.05$). **c** Representative flow cytometry plots: BM pre–proB (blue), proB (green), preB (orange) and control wild-type (black) B220+IgM−IgD− (left) with *lacZ* MFI (right). **d** *Erg* expression by RNA-seq (mean ± SD, Fragments Per kilobase of transcript per million mapped reads, FPKM) in *Erg*fl/fl pre-proB, proB and preB cells (n = 2 biologically independent samples) **e** *Erg* RNA-seq (FPKM) in *Erg*fl/fl and *Rag1Cre*T/+;*Erg*Δ/Δ pre-proB cells (n = 2 biologically independent samples; *edgeR two-sided adjusted *P* value for multiple comparisons = 1.41e−5, Supplementary Data 1) and *Erg* locus RNA-seq in *Erg*fl/fl (WT) and *Rag1Cre*T/+;*Erg*Δ/Δ (Erg KO, with pink highlighting absent expression) in pre–proB cells, H3K4me3 and H3k27ac ChIP-seq in wild-type proB cells and chromatin accessibility (ATAC-seq, blue). **f** *Erg*fl/fl (n = 4) and *Rag1Cre*T/+;*Erg*Δ/Δ (n = 7) biologically independent samples: B220+B-cell, Gr1+Mac1+ myeloid cell, and CD3+T-cell blood counts, mean ± SD; *P = 6.6e−8 by the Student's two-tailed unpaired *t* test (top left). B-lymphoid populations in *Erg*fl/fl (n = 9) and *Rag1Cre*T/+;*Erg*Δ/Δ (n = 10) biologically independent samples: BM as ratio of cell number to *Erg*fl/fl (mean ± SD, bottom left, see Supplementary Table 1). *$P_{adj}$ = 3.5e−6 (B220+), 3.5e−3 (B220+CD19−), 4.9e−10 (B220+CD19+), 3.0e−10 (IgM+IgD+), 3.5e−3 (IgM−IgD−), 1.7e−2 (pre-proB), 4.0e−2 (proB), 1.7e−2 (preB) by the Student's two-tailed unpaired *t* test corrected using the Holm's modification for multiple testing. Representative flow cytometry plots (right) with mean percentage of viable cells indicated. Source data are provided in the Source data file.

compared with *Erg*fl/fl and C57BL/6 controls (Fig. 2c). As these findings were also observed in *Pax5* deficient cells[13,23] reflecting a direct role for Pax5 in co-ordinating the structure of the *Igh* locus[14], we mapped *Erg* binding sites across the *Igh* locus by ChIP-seq. Unlike well-defined Pax5 binding to Pax5- and CTCF-associated intergenic regions (PAIR domains)[14,16], Erg binding to VH families was not identified across the locus (Fig. 2c, Supplementary Fig. 3a). Thus, a structural role for Erg in maintaining the multiple long-range interactions and VH-to-DJH recombination in normal cells is unlikely and cannot account for the absence of these in *Rag1Cre*T/+;*Erg*Δ/Δ pre–proB cells. Analysis of *Igh* locus accessibility by ATAC-

seq did not reveal any significant difference between Erg-deficient pre–proB, proB and preB cells and control cells (Supplementary Fig. 3a), suggesting that the loss of locus accessibility either by chromatin regulation[24] or by peripheral nuclear positioning with lamina-associated domain silencing[25] were not mechanisms that could adequately explain reduced *Igh* locus contraction, reduction of long range interactions, and loss of VH-to-DJH recombination in the absence of *Erg*.

A potential role for ETS family of transcription factors in regulation of immunoglobulin gene rearrangement was proposed from experiments investigating the iEµ enhancer: a complex *cis*-

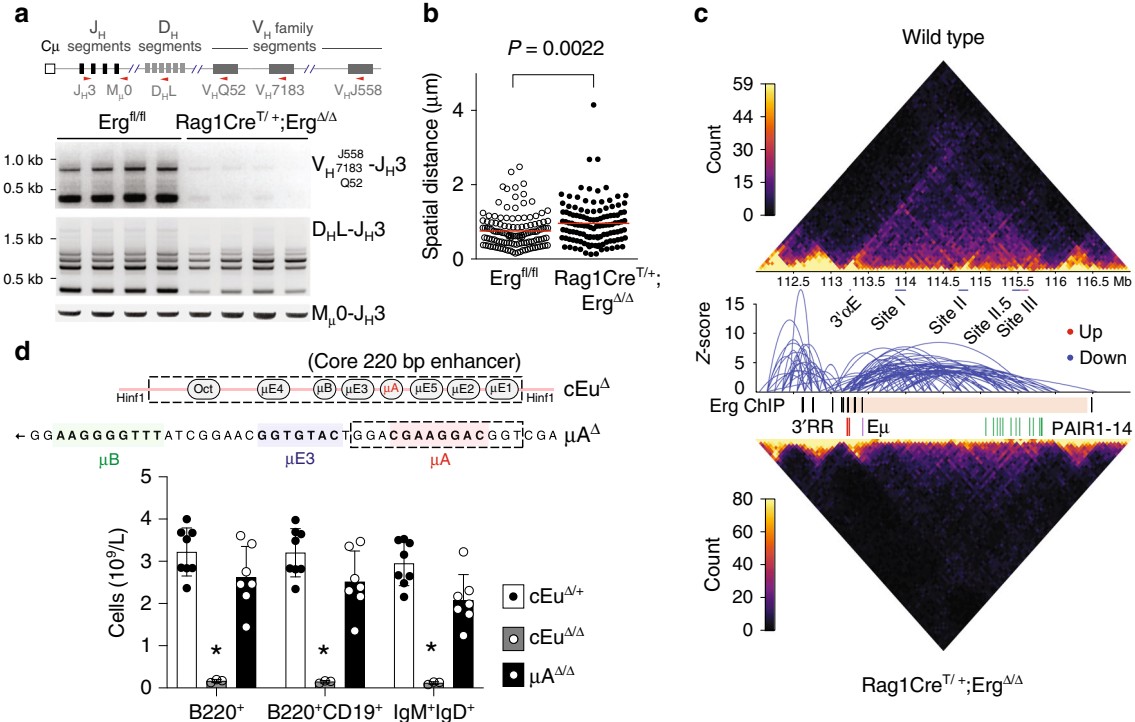

**Fig. 2 The immunoglobulin heavy chain locus in *Rag1Cre*$^{T/+}$;*Erg*$^{Δ/Δ}$ mice. a** Genomic PCR using degenerate primers (approximate locations indicated by red arrowheads) to *Igh* locus $V_H$J558, $V_H$7183, $V_H$Q52 segments for detection of $V_H$ to $DJ_H$ (top panel) and $D_H$ to $J_H$ (middle panel) recombination with Mu0 loading controls (bottom panel) in B220$^+$ BM cells. Representative of three independent experiments. **b** Intra-chromosomal distance between distal $V_H$J558 and proximal $V_H$7183 $V_H$ families by fluorescent in situ hybridisation from (n = 129) *Igh* alleles from *Erg*$^{fl/fl}$ and *Rag1Cre*$^{T/+}$;*Erg*$^{Δ/Δ}$ proB cells and pre–proB cells, respectively. *P* value by the Student's two-tailed unpaired *t* test. **c** Differential long-range chromatin interactions identified by high-throughput chromatin conformation capture analysis (in situ Hi-C) of the *Igh* locus between C57BL6 (wild-type) proB cells and *Rag1Cre*$^{T/+}$;*Erg*$^{Δ/Δ}$ pre–proB cells with interaction counts across the *Igh* locus shown. Reduced long-range interactions in *Rag1Cre*$^{T/+}$;*Erg*$^{Δ/Δ}$ B-cell progenitors are indicated by blue arcs. Erg binding by ChIP (black bars) across the heavy chain locus (pink shading) as indicated. The location of 3′regulatory region (3′RR, red bars), iEμ enhancer (Eμ, purple bar) and PAIR domains (green bars) are indicated. **d** Schematic representation of iEμ enhancer with the core 220 bp cEμ$^Δ$ deletion and μA$^Δ$ deletion shown (top). Peripheral blood counts of B220$^+$, B220$^+$CD19$^+$ and IgM$^+$IgD$^+$ cells in cEμ$^{Δ/+}$ (n = 8), cEμ$^{Δ/Δ}$ (n = 3) and μA$^{Δ/Δ}$ (n = 7) mice (bottom). *$P_{adj}$ value 2.6e−5 (B220$^+$), 1.8e−5 (B220$^+$CD19$^+$) and 9.0e−6 (IgM$^+$IgD$^+$) by the Student's two-tailed unpaired *t* test comparing cEμ$^{Δ/Δ}$ to cEμ$^{Δ/+}$ controls with the Benjamini Hochberg correction for multiple testing. Source data are provided in the Source data file.

activating element located in the intronic region between the *Igh* joining region ($J_H$) and constant region (Cμ) implicated in efficient $V_H$-to-$DJ_H$ recombination and *Igh* chain transcription[26]. The iEμ enhancer is proposed to nucleate a three-loop domain at the 3′ end of *Igh* interacting with the $V_H$ region to juxtapose 5′ and 3′ ends of the heavy chain locus[27]. Erg and its closest related ETS family member, Fli1, were shown to bind to the μA element and trans-activate iEμ co-operatively with a bHLH transcription factor in vitro[28]. We therefore sought to determine whether the lack of Erg, and Erg binding in particular to the μA site of iEμ, could account for the loss of $V_H$-to-$DJ_H$ recombination observed in *Rag1Cre*$^{T/+}$;*Erg*$^{Δ/Δ}$ mice in vivo. While ChIP-PCR demonstrated Erg binding to the iEμ enhancer containing the μA element (Supplementary Fig. 3b), mice in which the μA region (μA$^{Δ/Δ}$) was deleted had preserved numbers of circulating mature B cells compared with cEμ$^{Δ/+}$ controls (Fig. 2d) and intact $V_H$-to-$DJ_H$ recombination (Supplementary Fig. 3c). This was in contrast to cEμ$^{Δ/Δ}$ mice, in which a core 220 bp element of iEμ was deleted, in which a marked reduction of circulating mature IgM$^+$ IgD$^+$ B cells was evident in peripheral blood, in keeping with previous models[29] (Fig. 2d). Importantly, ChIP-seq did not demonstrate Erg binding to other iEμ enhancer regions in μA$^{Δ/Δ}$ proB cells (Supplementary Fig. 3d). Together these data show that-while Erg can bind to the μA region of the iEμ in vivo, deletion of this region did not result in significant perturbation of B lymphoid development. It is therefore unlikely that Erg binding to μA element

of iEμ could account for the loss of $V_H$-to-$DJ_H$ recombination in particular, or the *Rag1Cre*$^{T/+}$;*Erg*$^{Δ/Δ}$ phenotype in general.

**Rearranged *IgH* allele permits *Erg*-deficient B lymphopoiesis.** Given the loss of $V_H$-to-$DJ_H$ recombination associated with structural perturbation of the *Igh* locus in Erg-deficient pre–proB cells, we sought to complement the loss of formation of a functional *Igh* μ transcript and in doing so, determine whether failure to form a pre-BCR complex was a principal reason for the developmental block in *Rag1Cre*$^{T/+}$;*Erg*$^{Δ/Δ}$ mice[30]. Complementation with a functionally rearranged *Igh* allele in models of defective $V_H$-to-$DJ_H$ recombination such as deletion of *Rag1*, *Rag2*, or components of DNA-dependent protein kinase (DNA-PK) that mediate $V_H$-to-$DJ_H$ recombination, can overcome the pre-BCR developmental block[31–34].

The *IgH*$^{VH10tar}$ knock-in allele that expresses productive *Igh*$^{HEL}$ transcripts under endogenous *Igh* locus regulation[32] was therefore used to generate mice that lacked *Erg* in B-cell progenitors but would undergo stage-appropriate expression of the rearranged *Igh*$^{HEL}$ chain (*Rag1Cre*$^{T/+}$;*Erg* $^{Δ/Δ}$;*IgH*$^{VH10tar/+}$). The presence of the *IgH*$^{VH10tar}$ allele permitted B-cell development in the absence of *Erg*. The bone marrow of *Rag1Cre*$^{T/+}$;*Erg*$^{Δ/Δ}$;*IgH*$^{VH10tar/+}$ mice contained significant numbers of B220$^+$IgM$^+$ B cells and, notably, CD25$^+$CD19$^+$IgM$^-$ preB cells, a population coincident with successful pre-BCR formation[35], that were virtually absent in

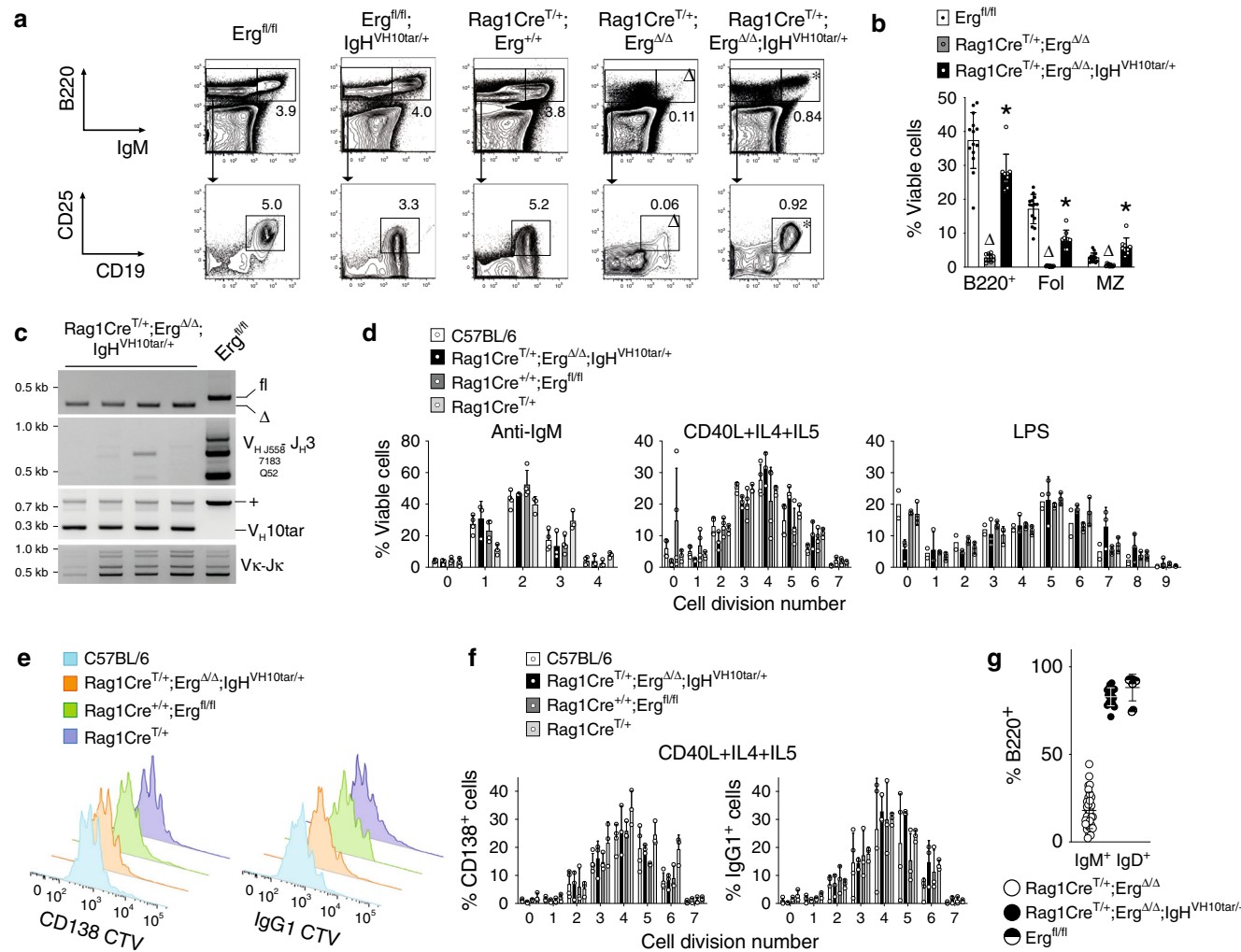

**Fig. 3 Rearranged *Igh* allele permits B-lymphoid development in the absence of Erg. a** Representative flow cytometry plots of BM B lymphoid populations (*n* = 9 *Erg^fl/fl^*, *n* = 8 *Erg^fl/fl^;IgH^VH10tar/+^*, *n* = 7 *Rag1Cre^T/+^*, *n* = 8 *Rag1Cre^T/+^;Erg^Δ/Δ^*, *n* = 8 *Rag1Cre^T/+^;Erg^Δ/Δ^;IgH^VH10tar/+^* biologically independent samples) with mean percentage of viable cells indicated; B220/IgM profile from whole BM, CD25/CD19 profile from B220^+^IgM^−^ BM cells. ^Δ^*P_adj_* = 2.1e−9 (B220^+^ IgM^+^) and 3.5e−4 (B220^+^IgM^−^CD25^+^CD19^+^) comparing *Rag1Cre^T/+^;Erg^Δ/Δ^* to *Erg^fl/fl^*; *\*P_adj_* = 3.1e−3 (B220^+^IgM^+^) and 3.3e−2 (B220^+^IgM^−^CD25^+^ CD19^+^) *Rag1Cre^T/+^;Erg^Δ/Δ^;IgH^VH10tar/+^* to *Rag1Cre^T/+^;Erg^Δ/Δ^*. **b** Proportions of viable splenic B lymphoid populations (mean ± SD; *n* = 14 *Erg^fl/fl^*, *n* = 10 *Rag1Cre^T/+^;Erg^Δ/Δ^*, *n* = 9 *Rag1Cre^T/+^;Erg^Δ/Δ^;IgH^VH10tar/+^* biologically independent samples). ^Δ^*P_adj_* = 2.1e−11 (B220^+^), 5.6e−11 (Fol), 3.3e−5 (MZ) comparing *Erg^fl/fl^* to *Rag1Cre^T/+^;Erg^Δ/Δ^*; *\*P_adj_* = 2.1e−10 (B220^+^), 3.2e−8 (Fol), 3.2e−6 (MZ) comparing *Rag1Cre^T/+^;Erg^Δ/Δ^* to *Rag1Cre^T/+^;Erg^Δ/Δ^*; *IgH^VH10tar/+^* by the Student's two-tailed unpaired *t* test corrected using the Holm's modification for multiple testing. Fol follicular, MZ marginal zone (see Supplementary Fig. 1, Supplementary Table 1). **c** PCR of genomic DNA from B220^+^ splenocytes for *Erg* (top panel; fl, floxed allele, Δ, cre-deleted allele), V_H_-to-DJ_H_ recombination of V_HJ558_, V_H7183_, V_HQ52_ families (second panel), V_H_10tar allele (third panel) and V_κ_ light chain recombination (bottom panel). Representative of three independent experiments. **d** Proliferation by cell trace violet assay of wild type (C57BL/6), *Rag1Cre^T/+^;Erg^Δ/Δ^;IgH^VH10tar/+^*, *Rag1Cre^+/+^;Erg^fl/fl^* and *Rag1Cre^T/+^* B220^+^ splenocytes stimulated with anti-IgM, CD40L + IL4 + IL5 (T-cell-dependent) or LPS (T-cell-independent) stimulation. Mean percentage of viable cells ± SD for each cell division shown. No significant differences between genotypes were observed (*P* > 0.90, two-way ANOVA). **e** Representative flow cytometry plots showing CD138^+^ differentiation and IgG1 class switch by of B220^+^ splenocytes in response to CD40L + IL4 + IL5 stimulation using CTV labelling (blue, C57BL6; orange, *Rag1Cre^T/+^;Erg^Δ/Δ^;IgH^VH10tar/+^*; green, *Erg^fl/fl^*; purple, *Rag1Cre^T/+^*). **f** CD138^+^ differentiation and IgG1 class switch of B220^+^ cells by cell division. Mean percentage of viable cells ± SD for each cell division shown. No significant differences were observed between genotypes (*P* > 0.99, two-way ANOVA for both CD138^+^ cell differentiation and IgG1 switch). For **d**, **e**, **f** *n* = 2–4 C57BL/6, *n* = 5-6 *Rag1Cre^T/+^;Erg^Δ/Δ^;IgH^VH10tar/+^*, *n* = 3–5 *Rag1Cre^+/+^;Erg^fl/fl^* and *n* = 3 *Rag1Cre^T/+^* mice. **g** Percentage (mean ± SD) circulating IgM^+^IgD^+^ B220^+^ B cells in *Rag1Cre^T/+^;Erg^Δ/Δ^* (*n* = 31), *Rag1Cre^T/+^;Erg^Δ/Δ^;IgH^VH10tar/+^* (*n* = 17), and *Erg^fl/fl^* (*n* = 9) mice. *P* = 1.06e−27 comparing *Rag1Cre^T/+^;Erg^Δ/Δ^* to *Rag1Cre^T/+^;Erg^Δ/Δ^;IgH^VH10tar/+^* by the Student's two-tailed unpaired *t* test. Source data are provided in the Source data file.

*Rag1Cre*^T/+^;*Erg*^Δ/Δ^ mice (Fig. 3a). The rescued preB cells, however, were at lower numbers compared with *Erg*^fl/fl^, *Erg*^fl/fl^;*IgH*^VH10tar^ and *Rag1Cre*^T/+^ controls. This was likely due to restriction of the clonal repertoire permitted by the *IgH*^VH10tar^ allele as the predominant *Igh* clone (Fig. 3c).

Similarly, in the spleens of *Rag1Cre*^T/+^;*Erg*^Δ/Δ^;*IgH*^VH10tar/+^ mice, near normal numbers of all B-lymphoid populations were

observed, in contrast to the marked reduction in *Rag1Cre*^T/+^; *Erg*^Δ/Δ^ mice (Fig. 3b). Notably, IgκL chain recombination had proceeded in *Rag1Cre*^T/+^;*Erg*^Δ/Δ^;*IgH*^VH10tar/+^ cells (Fig. 3c).

We next tested whether the rescued *Rag1Cre*^T/+^;*Erg* ^Δ/Δ^; *IgH*^VH10tar/+^ splenic B cells were functional in the absence of *Erg*. *Rag1Cre*^T/+^;*Erg*^Δ/Δ^;*IgH*^VH10tar/+^ splenocytes were indistinguishable from wild-type controls in in vitro proliferative assays using

anti-μ stimulation, T-cell dependent stimulation with CD40 ligand, IL4 and IL5, or T-cell independent stimulation using lipopolysaccharide (Fig. 3d). $Rag1Cre^{T/+};Erg^{\Delta/\Delta};IgH^{VH10tar/+}$ splenic B cells were also able to differentiate normally as measured by formation of CD138$^+$ plasma cells and IgG1 class switch recombination (Fig. 3e, f). Circulating $Rag1Cre^{T/+};Erg^{\Delta/\Delta};$ $IgH^{VH10tar/+}$ B cells also expressed IgD, unlike their $Rag1Cre^{T/+};$ $Erg^{\Delta/\Delta}$ counterparts (Fig. 3g). These experiments demonstrated that loss of a functional $Igh$ μ transcript and failure to form a pre-BCR complex was a principal reason for lack of B-cell development in $Rag1Cre^{T/+};Erg^{\Delta/\Delta}$ mice.

**Erg-deficient pre–proB cells do not express Ebf1 and Pax5.** To define the mechanism by which $Erg$ regulates $V_H$-to-$DJ_H$ recombination and pre-BCR formation, we undertook gene expression profiling of $Rag1Cre^{T/+};Erg^{\Delta/\Delta}$ pre–proB cells. Differential gene expression and gene-ontogeny analysis of differentially expressed genes in $Rag1Cre^{T/+};Erg^{\Delta/\Delta}$ pre–proB compared with $Erg^{fl/fl}$ pre–proB cells demonstrated deregulated expression of multiple B lymphoid genes (Fig. 4a). These included genes encoding cell surface or adhesion receptors and core components of the pre-BCR complex CD19, CD22, Igll1, Vpreb1, Vpreb2, CD79a and CD79b, genes required for $Igh$ recombination such as $Rag1$ and $Rag2$ and components of non-homologous end-joining repair complex associated with V(D)J recombination: $Xrcc6$ (Ku70) and $Lig4$, and importantly, transcription factors implicated in B-cell development ($Ebf1, Pax5, Tcf3, Bach2, Irf4, Myc, Pou2af1, Lef1, Myb$) (Fig. 4b).

$Ebf1$ and $Pax5$ are critical for B lineage specification[5] and maintenance[36,37] and act co-operatively to regulate a gene network in early B-cell fates[9]. Because we observed with the loss of $Erg$, reduced expression of several critical B lineage genes previously identified to be controlled by $Ebf1$ and/or $Pax5$, for example $CD19, Vpreb1,$ and $Igll1$ (Fig. 4a), we speculated that Erg may play an important role in regulating the expression of these two essential transcription factors and their targets. To determine if Erg bound $Ebf1$ and/or $Pax5$ gene regulatory regions and directly regulated their expression, we undertook ChIP-seq analysis in wild-type proB cells and ATAC-seq to assess locus accessibility at the $Ebf1$ and $Pax5$ loci in the absence of $Erg$ in $Rag1Cre^{T/+};Erg^{\Delta/\Delta}$ pre–proB cells and proB and preB cells rescued with the $IgH^{VH10tar}$ knock-in allele. This demonstrated direct Erg binding to the proximal (β) promoter region of $Ebf1$[38] as well as to the $Pax5$ promoter and $Pax5$ lymphoid-specific intron 5 enhancer[12] (Fig. 4c, Supplementary Fig. 4b). Direct Erg binding to these regulatory regions together with the absence of $Ebf1$ and $Pax5$ transcription in $Rag1Cre^{T/+};Erg^{\Delta/\Delta}$ pre–proB cells and the loss of Ebf1 and Pax5 protein in $Rag1Cre^{T/+};Erg^{\Delta/\Delta}$ by western blot (Fig. 4d), demonstrated that Erg was a direct transcriptional regulator of $Ebf1$ and $Pax5$. Importantly, the loss of $Ebf1$ and $Pax5$ expression occurred while expression of other known regulators of $Ebf1$ expression, namely, $Foxo1, Spi1, Tcf3$ and $Ikzf1$ were maintained (Supplementary Fig. 4a), and both $Ebf1$ and $Pax5$ loci remained accessible by ATAC-seq in $Rag1Cre^{T/+};Erg^{\Delta/\Delta}$ pre–proB cells (Fig. 4c). Reinforcing the observation that Erg, Ebf1 and Pax5 may form a co-ordinated transcriptional network, the $Erg$ promoter region was directly bound by Pax5, and the $Erg$ enhancer region was bound by Pax5 and Ebf1 (Fig. 4c).

To better understand the roles of Erg, Ebf1 and Pax5 in the B-cell lineage trajectory, single-cell RNA-seq of CLP, pre–proB and CD19$^+$ proB and preB populations was examined (GSE 114793, Fig. 4e). Consistent with our other analysis (Fig. 1b), an increase in $Erg$ expression in CLPs, pre–proB and proB cells was observed (Fig. 4e) with the identity of proB and preB populations

confirmed with analysis of additional B lineage genes (Supplementary Fig. 5). Importantly, $Erg$ expression preceded the expression of $Ebf1$ and $Pax5$ in the B lineage trajectory, with $Ebf1$ and $Pax5$ expression increasing during the later proB and preB stages. Taken together, this data strongly supported an apical role for Erg in initiating $Ebf1$ and $Pax5$ expression during early B-cell development.

**An $Erg$, $Ebf1$ and $Pax5$ co-dependent gene regulatory network.** As we observed Ebf1 and Pax5 binding to $cis$-regulatory regions of the $Erg$ locus (Fig. 4c), we determined whether Ebf1 and Pax5 could regulate $Erg$ gene expression in B-cell progenitors by examining a publicly available dataset in which $Ebf1$ ($Ebf1^{\Delta/\Delta}$) or $Pax5$ ($Pax5^{\Delta/\Delta}$) had been deleted (Fig. 5a). Deletion of either $Ebf1$ or $Pax5$ resulted in reduced $Erg$ expression (Fig. 5b), with Ebf1 appearing to be the stronger influence. We next compared gene expression changes in $Ebf1^{\Delta/\Delta}$ pre–proB cells and $Pax5^{\Delta/\Delta}$ proB cells to those genes regulated by Erg in pre–proB cells. As would be predicted if Erg, Ebf1 and Pax5 were components of a co-dependent gene regulatory network, this analysis showed a highly significant correlation in gene expression changes observed with $Ebf1$ or $Pax5$ deletion in pre–proB and proB cells and those observed with $Erg$ deletion in pre–proB cells. This was noted for downregulated genes in Erg, Ebf1 and Pax5 deficient cells in particular (Fig. 5c).

Finally, to confirm that Ebf1 and Pax5 were transcriptional regulators downstream of Erg, transduction of $Rag1Cre^{T/+};Erg^{\Delta/\Delta}$ pre–proB cells with MSCV-driven constructs for constitutive overexpression of $Ebf1$ and $Pax5$ was performed. This experiment demonstrated rescue of B220 expression with $Ebf1$ or $Pax5$ overexpression in Erg deficient cells (Fig. 5d). Notably, partial rescue of CD19 expression and $V_H$-to-$DJ_H$ recombination was observed with $Ebf1$ overexpression, while no rescue was observed with $Pax5$ overexpression (Fig. 5d, e). RNA-seq analysis of Erg-deficient cells transduced with $Ebf1$ or $Pax5$ expression vectors demonstrated $Ebf1$ overexpression could rescue the expression of several target genes of the transcriptional network including $Pax5$ itself, genes involved in pre-BCR signalling (such as $Vpreb1, Vpreb2, CD79a, CD79b, CD22$ and $CD19$), genes involved in V-to-$DJ_H$ recombination (such as $Rag1, Rag2$), as well as transcription from the $Igh$ locus ($Ighv1-5, Ighv1-7, Ighv1-4$). In the absence of $Ebf1$, $Pax5$ overexpression alone induced the expression of a much more limited set of these target genes (Fig. 5f). Therefore these data suggest that $Pax5$ lies downstream of $Ebf1$ and supports the model where Ebf1 facilitates the role of Pax5 in B-cell development[11]. These findings were also in keeping with a hierarchical model of Erg, Ebf1 and Pax5 forming a co-dependent transcriptional network that co-regulate critical target genes required for $V_H$-to-$DJ_H$ recombination and pre-BCR signaling.

**Erg co-binds common Ebf1 and Pax5 target genes.** Because expression of multiple B-cell genes were deregulated in $Rag1Cre^{T/+};Erg^{\Delta/\Delta}$ pre–proB cells, including those to which Ebf1 and Pax5 had been shown to directly bind and regulate, we investigated the possibility that Erg co-bound common target genes to reinforce the Ebf1 and Pax5 gene network using a genome-wide motif analysis of Erg DNA-binding sites in proB cells. As expected, the most highly enriched motif underlying Erg binding was the ETS motif. However, significant enrichment of $Ebf1-$, $E2A-$, $Pax5-$ and $Foxo1-$binding motifs were also identified within 50 bp of Erg-binding sites (Fig. 6a), suggesting that Erg acts co-operatively with other transcription factors to regulate target gene expression in a co-dependent gene network. Analysis of the binding of each of Erg, Ebf1 and Pax5 to

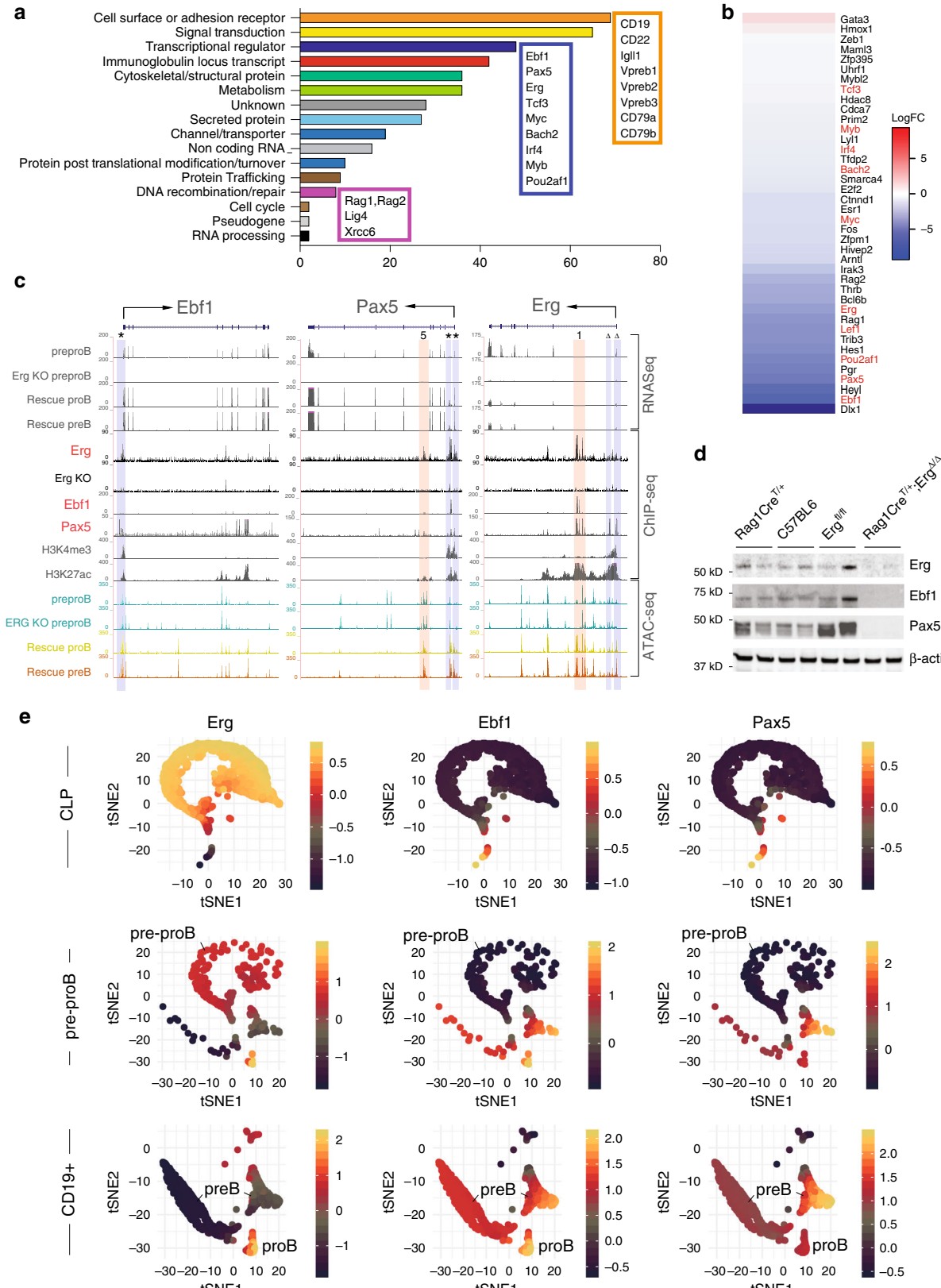

regulatory regions of genes that were differentially expressed in $Rag1Cre^{T/+};Erg^{\Delta/\Delta}$ pre–proB cells was then undertaken. This analysis identified significant overlap of Erg-, Ebf1- and Pax5-binding sites within 5 kb of the transcriptional start site (TSS) of genes differentially expressed in $Rag1Cre^{T/+};Erg^{\Delta/\Delta}$ pre–proB cells compared with control pre–proB cells (Fig. 6b). Taken

together, these data provided further compelling evidence for a gene regulatory network in which Erg is required for initiating and maintaining expression of *Ebf1* and *Pax5* from the pre–proB cell stage of development, as well as reinforcing expression of target genes within the network by co-operative binding and co-regulation of target genes with Ebf1 and Pax5.

**Fig. 4 Gene expression in $Rag1Cre^{T/+};Erg^{\Delta/\Delta}$ pre–proB cells and Erg DNA binding. a** Differentially expressed genes in $Rag1Cre^{T/+};Erg^{\Delta/\Delta}$ pre–proB cells compared with $Erg^{fl/fl}$ controls, manually curated according to function based on GO term analysis (see Supplementary Data 1) with the number of genes for each functional category shown by the horizontal axis and selected genes highlighted in boxes. **b** Differential expression of transcription factors[77] between $Erg^{fl/fl}$ and $Rag1Cre^{T/+};Erg^{\Delta/\Delta}$ pre–proB cells, ordered by logFC, with selected B lineage transcription factors highlighted in red. **c** RNA-seq for $Ebf1$, $Pax5$ and $Erg$ loci, with ChIP-seq for Erg binding in C57BL/6 proB cells and thymic $Rag1Cre^{T/+};Erg^{\Delta/\Delta}$ Erg knockout cells (Erg KO) to control for sites of non-Erg ChIP binding to DNA (see also Supplementary Fig. 4b). Ebf1, Pax5, H3K4me3 promoter mark, H3K27ac promoter and enhancer mark in proB cells by ChIP-seq, and ATAC-seq in $Erg^{fl/fl}$ pre–proB cells (pre–proB), $Rag1Cre^{T/+};Erg^{\Delta/\Delta}$ pre–proB (Erg KO pre–proB), and Erg-deficient proB (Rescue proB) and preB (Rescue preB) cells in $Rag1Cre^{T/+};Erg^{\Delta/\Delta};IgH^{VH10tar/+}$ mice that develop with a functionally rearranged $Igh$ allele as shown. The asterisk (*) indicates Erg binding to the promoter region (shaded blue) of $Ebf1$ and $Pax5$. Erg binding to intragenic enhancer regions of $Pax5$ (shaded pink) with intron number as indicated. The delta ($\Delta$) indicates Pax5 binding to $Erg$ promoter region. Ebf1 and Pax5 binding to $Erg$ intragenic enhancer regions (shaded pink) with intron number as indicated (see also Supplementary Fig. 4b). **d** Western blot for Erg, Ebf1, Pax5 and β-actin in $Rag1Cre^{T/+}$, C57BL/6 and $Erg^{fl/fl}$ proB and $Rag1Cre^{T/+};Erg^{\Delta/\Delta}$ pre–proB cells (two samples of each genotype are shown). Representative of two independent experiments. **e** Single-cell RNA-seq analysis (3297 cells, GSE114793) showing t-distributed stochastic neighbour embedding (tSNE) plots of CLP, pre–proB and CD19$^+$ proB and preB populations demonstrating the imputed expression of $Erg$, $Ebf1$ and $Pax5$ in single cells in the B lineage trajectory (see also Supplementary Fig. 5). Source data are provided in the Source data file.

To further delineate the directly regulated target genes in an Erg-dependent Ebf1 and Pax5 transcriptional network, we undertook mapping of ChIP-seq binding of Erg, Ebf1 and Pax5 to Erg-dependent genes at the pre–proB cell stage of development. The majority of these target genes demonstrated direct combinatorial binding of Erg, Ebf1 and/or Pax5 to annotated promoter regions, gene body enhancer/putative enhancer regions or putative distal enhancer regions of these genes (Fig. 6c). Detailed examination of several key target genes for which expression was completely dependent on Erg in pre–proB cells identified direct binding of Erg to the promoter and enhancer regions for several pre-BCR components, including $CD19$, $Igll1$, $Vpreb1$ and $CD79a$. This occurred with co-ordinate binding of Ebf1 and Pax5 to the regulatory regions of these genes[15] (Fig. 6d). In addition, indirect regulation by Erg at the $Rag1/Rag2$ locus was also identified, with down-regulation of expression of transcription factors that bind and regulate the $Rag2$ promoter such $Pax5$, $Lef1$ and $c$-$Myb$ in $Rag1Cre^{T/+};Erg^{\Delta/\Delta}$ pre–proB cells (Fig. 4b)[39], as well as direct binding of Erg to the conserved B-cell specific $Erag$ enhancer[40] (Supplementary Fig. 4a, b). Importantly, the loss of $Rag1$ and $Rag2$ expression in $Rag1Cre^{T/+};Erg^{\Delta/\Delta}$ pre–proB cells occurred while expression of $Foxo1$, a positive regulator of the locus[41] was relatively maintained (Supplementary Fig. 4a).

An Erg-Ebf1-Pax5 mediated gene regulatory network was then mapped using each target gene, expression of which was perturbed in $Rag1Cre^{T/+};Erg^{\Delta/\Delta}$ pre–proB cells, and that was directly bound by Erg, Ebf1 and/or Pax5 at promoter, proximal or distal gene regions, to provide a comprehensive representation of this gene network (Fig. 6e). This highlights the interdependent roles of these transcription factors in multiple cellular processes required for B lymphopoiesis.

An important observation arising from our data was that the B-lymphoid developmental block arising in $Rag1Cre^{T/+};Erg^{\Delta/\Delta}$ pre–proB cells could be overcome with the provision of a rearranged functional $Igh$ VH10tar allele. This suggested that once the pre-BCR checkpoint was bypassed, $Erg$ was no longer critical for further B-cell development and function, including $V_L J_L$ recombination of the $Igl$ and BCR formation (Fig. 3c, d). Indeed, beyond the pre-BCR checkpoint, re-emergence of $Ebf1$ and $Pax5$ expression occurred (Fig. 4c) as well as expression of target genes of the $Ebf1$ and $Pax5$ network (Fig. 6d, Supplementary Fig. 4a) in Erg-deficient $Rag1Cre^{T/+};Erg^{\Delta/\Delta};IgH^{VH10tar/+}$ proB and preB cells rescued with a VH10tar allele. This was in keeping with the expression pattern of $Erg$ in the B lineage trajectory (Figs. 1b–d and 4e) and defines the role of Erg as an exquisitely stage-specific regulator of early B-cell development.

## Discussion

In this study we explored the role of the transcription factor Erg in B lymphopoiesis. Our studies suggest that $Erg$ expression from the CLP stage of development initiates a transcriptional network comprised of Erg, Ebf1 and Pax5 in pre–proB and proB cells to regulate $V_H$-to-$DJ_H$ $Igh$ recombination and pre-BCR signalling (Fig. 1b, Fig. 4e).

This important role for Erg in B-cell development was demonstrated in mice in which $Erg$ had been deleted throughout lymphopoiesis, which exhibited a developmental block at the pre–proB cell stage that was associated with profound defects in $V_H$-to-$DJ_H$ recombination, $Igh$ locus organisation and transcriptional changes in multiple B-cell genes, including loss of expression of $Ebf1$, and $Pax5$. Combining RNA-seq, ChIP-seq and gene complementation studies, we were able to define a co-dependent transcriptional network between Erg, Ebf1 and Pax5, with direct Erg binding to the proximal (β) $Ebf1$ promoter, to which Pax5, Ets1 and Pu.1 also co-operatively bind[38], as well as Erg binding to the $Pax5$ promoter and potent intron 5 enhancer region, two critical regulatory elements required for correct transcriptional initiation of $Pax5$ in early B-cell development[12]. These data support a model (Fig. 6f) in which increased $Erg$ expression from CLPs is required to initiate and maintain $Ebf1$ and $Pax5$ expression in pre–proB cells and proB cells, to establish an inter-dependent B-lymphoid gene regulatory network.

Together Erg, Ebf1 and Pax5 directly co-regulated the expression of multiple genes that had previously been identified as direct transcriptional targets of Ebf1 and Pax5 (Fig. 6c–e). Direct Erg binding to promoters of the pre-BCR signalling complex genes such as $Igll1$, $VpreB$ and $CD79a$, establish Erg as a transcriptional regulator of target genes in this network. In addition to $Rag1$ and $Rag2$, we also identified network regulation of expression of $Xrcc6$, the gene encoding the Ku70 subunit of DNA-dependent protein kinase holoenzyme (DNA-PK) that binds DNA double strand breaks during V(D)J recombination[42], and $Lig4$, encoding the XRCC4-associated DNA-ligase that is required for DNA-end joining during V(D)J recombination[43] (Supplementary Fig. 4a, b). Along with direct Erg promotion of expression of $Pax5$, a structural regulator of the $Igh$ locus, these findings are sufficient to explain the $Rag1Cre^{T/+};Erg^{\Delta/\Delta}$ pheno-type in which $V_H$-to-$DJ_H$ recombination was lost. Together with loss of expression of components of the pre-BCR complex, we can conclude B-cell development was blocked as a consequence of $Erg$ deletion due to the collapse of the Erg-mediated transcriptional network.

Importantly, re-emergence of $Ebf1$ and $Pax5$ expression beyond the pre-BCR checkpoint in $Igh$-rescued $Rag1Cre^{T/+};Erg^{\Delta/\Delta};IgH^{VH10tar/+}$ proB and preB cells was observed, along with

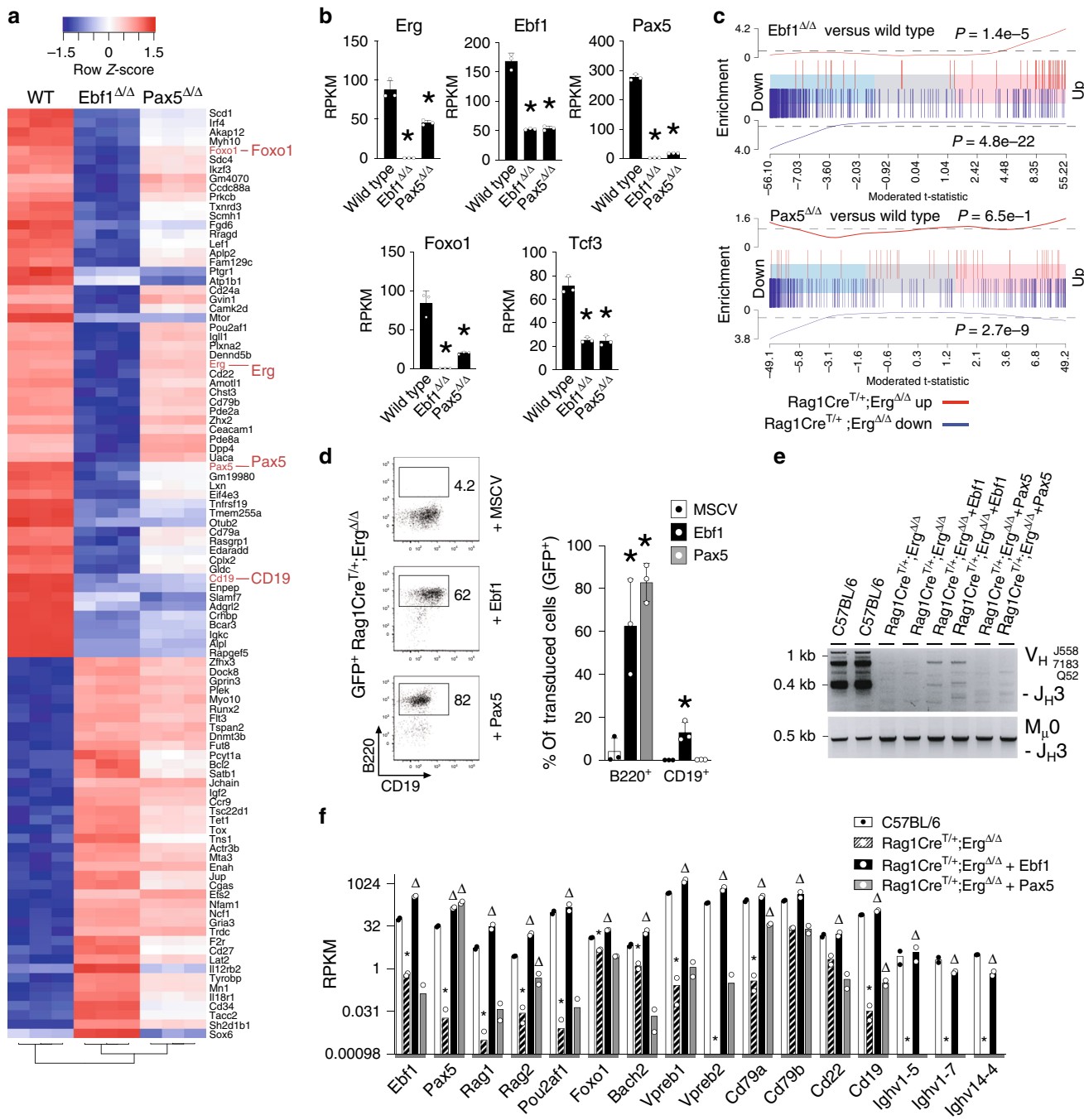

expression of target genes of Ebf1 and Pax5. This demonstrates that Erg is a stage-specific regulator of early B-cell development, with emergence of an Erg-independent Ebf1 and Pax5 gene network during the later stages of B-cell development, once clones have transitioned through the pre-BCR checkpoint. This would allow IgL chain $V_L$ to $J_L$ recombination and BCR formation to proceed in preB cells in which endogenous *Erg* expression is also reduced (Figs. 1b, c and 4e). The transcriptional regulators of *Ebf1* and *Pax5* expression during these later stages of B-cell development remain to be defined.

*Erg*, however, is critical for initiating and maintaining *Ebf1* and *Pax5* expression in pre–proB and proB cells (Fig. 4e), orchestrating a transcriptional network required for early B-cell development. In this role, Erg not only co-ordinates the transcriptional functions of Ebf1 and Pax5, but also directly binds and activates

critical target genes required for transition through the pre-BCR checkpoint.

## Methods
**Mice.** Mice carrying the *Erg*tm1a(KOMP)wtsi knock-first reporter allele[44] (*Erg*KI, KOMP Knockout Mouse Project) were generated by gene targeting in ES cells. Mice with a conditional *Erg* knockout allele (*Erg*fl) from which the IRES-LacZ cassette was excised were generated by interbreeding *Erg*KI mice with *Flpe* transgenic mice[45]. *Rag1Cre* mice[46], in which Cre recombinase is expressed during lymphopoiesis from the CLP stage[22], were interbred with *Erg*fl mice to generate mice lacking Erg in lymphopoiesis (*Rag1Cre*T/+;*Erg*Δ/Δ) and *Rag1Cre*+/+;*Erg*fl/fl (*Erg*fl/fl) controls. Mice carrying the rearranged immunoglobulin heavy chain *IgH*VH10tar allele[47] were a gift from Professor Robert Brink. The cEμΔ/Δ and μAΔ/Δ mice were generated by the MAGEC laboratory (Walter and Eliza Hall Institute of Medical Research)[48] on a C57BL/6J background. To generate cEμΔ mice, 20 ng/μl of Cas9 mRNA, 10 ng/μl of sgRNA (GTTGAGGATTCAGCCGAAAC and ATGTTGAGTTGGAGTCAAGA) and 40 ng/μl of oligo donor

**Fig. 5 Gene expression in *Ebf1*- and *Pax5*-deficient cells and rescue of *Erg*-deficient pre–proB cells. a** Heatmap of top 100 most variable genes in wild-type (WT, $n = 3$) proB, $Ebf1^{\Delta/\Delta}$ ($n = 3$) pre–proB, $Pax5^{\Delta/\Delta}$ ($n = 3$) proB cells with hierarchical clustering applied. **b** Expression of *Erg*, *Ebf1*, *Pax5*, *Foxo1* and *Tcf3* in wild-type ($n = 3$), $Ebf1^{\Delta/\Delta}$ ($n = 3$) and $Pax5^{\Delta/\Delta}$ ($n = 3$) cells (RPKM, mean ± SD shown). *Compared with wild type by the Student's two-tailed unpaired $t$ test, *Erg*: $P = 2.56e{-}4$ and $4.65e{-}3$; *Ebf1* expression: $P = 1.28e{-}4$ and $1.48e{-}4$; *Pax5* expression: $P = 1.32e{-}6$ and $1.68e{-}6$; *Foxo1* expression: $P = 7.46e{-}4$ and $2.09e{-}3$; and *Tcf3* expression: $P = 5.01e{-}4$ and $6.85e{-}4$ in $Ebf1^{\Delta/\Delta}$ and $Pax5^{\Delta/\Delta}$ cells respectively. **c** Barcode enrichment plots depicting strongly associated gene expression signatures of down (vertical blue bars) and up (vertical red bars) regulated genes in $Rag1Cre^{T/+};Erg^{\Delta/\Delta}$ pre–proB cells compared with $Ebf1^{\Delta/\Delta}$ pre–proB (top) and $Pax5^{\Delta/\Delta}$ proB (bottom) cells. Genes are ordered (from left to right) as most downregulated to most upregulated in $Ebf1^{\Delta/\Delta}$ or $Pax5^{\Delta/\Delta}$ cells. X-axis: moderated t-statistic in $Ebf1^{\Delta/\Delta}$ or $Pax5^{\Delta/\Delta}$ versus wild-type. A camera gene set test[55] confirmed the correlation, $P$ values are shown for up- and downregulated genes. **d** Representative flow cytometry plots and percentage (Mean ± SD) of GFP$^+$ B220$^+$ and CD19$^+$ cells from lineage-negative $Rag1Cre^{T/+};Erg^{\Delta/\Delta}$ BM transduced with MSCV control ($n = 3$), MSCV-Ebf1 ($n = 3$) or MSCV-Pax5 ($n = 3$) GFP$^+$ retroviruses cultured on OP9 stromal cells with IL-7, SCF and Flt3-ligand. *$P_{adj} = 1.14e{-}2$ for MSCV-Ebf1 and $2.20e{-}4$ for MSCV-Pax5 (GFP$^+$B220$^+$), and $8.86e{-}3$ for MSCV-Ebf1 (GFP$^+$CD19$^+$), Student's two-tailed unpaired $t$ test with Holm's modification for multiple testing, compared with MSCV. **e** V$_H$-to-DJ$_H$ recombination of V$_H$J558, V$_H$7183, V$_H$Q52 segments (top panel) with Mu0 loading controls (bottom panel) in B220$^+$-enriched lineage−negative C57BL/6 ($n = 2$), $Rag1Cre^{T/+};Erg^{\Delta/\Delta}$ ($n = 2$), and $Rag1Cre^{T/+};Erg^{\Delta/\Delta}$ BM transduced with MSCV-Ebf1 ($n = 2$) or MSCV-Pax5 ($n = 2$) retroviruses. **f** Expression of selected B-cell genes (RPKM) by RNA-seq in B220$^+$ C57BL/6 proB cells ($n = 2$), $Rag1Cre^{T/+};Erg^{\Delta/\Delta}$ pre–proB cells ($n = 2$), and $Rag1Cre^{T/+};Erg^{\Delta/\Delta}$ cells transduced with MSCV-Ebf1 ($n = 2$) or MSCV-Pax5 ($n = 2$) retroviruses. Limma two-sided adjusted $P$ value: *$P_{adj} < 0.05$ compared with C57BL/6; $^{\Delta}P_{adj} < 0.05$ compared with $Rag1Cre^{T/+};Erg^{\Delta/\Delta}$. See Source data file for individual $P_{adj}$ values. $n$ number of biologically independent samples. Source data are provided in the Source data file.

(CAAGCTAAAATTAAAAGGTTGAACTCAATAAGTTAAAAGAGGACCTCTC CAGTTTCGGCTCAACTCAACATTGCTCAATTCATTTAAAAATATTTGAA ACTTAATTTATTATTGTTAAAA) were injected into the cytoplasm of fertilised one-cell stage embryos. To generate μA$^{\Delta}$ mice, 20 ng/μl of Cas9 mRNA, 10 ng/μl of sgRNA (GAACACCTGCAGCAGCTGGC) and 40 ng/μl of oligo donor (GCTACA AGTTTACCTAGTGGTTTTATTTTCCCTTCCCCAAATAGCCTTGCCACATG ACCTGCCAGCTGCTGCAGGTGTTCTGGTTCTGATCGGCCATCTTGACTCC AACTCAACATTGCT) were injected into the cytoplasm of fertilized one-cell stage embryos. Twenty-four hours later, two-cell stage embryos were transferred into the oviducts of pseudo-pregnant female mice. Viable offspring were genotyped by next-generation sequencing. Non-commercial unique materials are subject to Materials Transfer Agreements. Mice were co-housed in a barrier facility and analysed from 6 to 18 weeks of age. Male and female mice were used. The primers and PCR conditions used for genotyping are provided in Supplementary Table 3. This study was performed in accordance with the Australian Code for the Care and Use of Animals for Scientific Purposes, published by the Australian National Health and Medical Research Council. Euthanasia was performed by $CO_2$ induction or cervical dislocation. Experimental procedures were approved by the Walter and Eliza Hall Institute of Medical Research Animal Ethics Committee.

**Primary cell culture**. B-cell progenitors were obtained from bone marrow that was lineage depleted using biotinylated Ter119, Mac1, Gr1, CD3, CD4, and CD8 antibodies, anti-biotin microbeads and LS columns (Miltenyi Biotec) and cultured on OP9 stromal cells in Iscove's Modified Dulbecco's Medium (Gibco, Invitrogen) supplemented with 10% (v/v) foetal calf serum (Gibco, Invitrogen), 50 μM β-mercaptoethanol as well as murine interleukin-7 (10 ng/mL) at 37 °C in 10% $CO_2$ for 7 days. Splenic B cells were purified by negative selection using a B-cell isolation kit (Miltenyi Biotec)[49] and purity was confirmed by flow cytometry prior to labelling with Cell Trace Violet (CTV; Life technologies) as per the manufacturer's instructions. Labelled cells were seeded at $5 \times 10^4$ cells per well and cultured for 90 h.

**Haematology**. Blood was collected into tubes containing EDTA (Sarstedt) and analysed on an Advia 2120 analyser (Bayer).

**Flow cytometry**. Single-cell suspensions from bone marrow, lymph node or spleen were prepared in a balanced salt solution (BSS-CS: 0.15 M NaCl, 4 mM KCl, 2 mM CaCl$_2$, 1 mM MgSO$_4$, 1 mM KH$_2$PO$_4$, 0.8 mM K$_2$HPO$_4$, and 15 mM HEPES supplemented with 2% [vol/vol] bovine calf serum). Analysis of blood was performed after erythrocyte lysis in buffered 156 mM NH$_4$Cl. Staining was performed using biotinylated or fluorochrome-conjugated antibodies specific for murine antigens Ter119 (Ly-76), CD41 (MWReg30), Gr1 (Ly6G and Ly6C), Mac1 (CD11b), NK1.1, CD11c (N418), CD45R/B220 (RA3-6B2), CD19 (1D3), CD3 (17A2), CD4 (GK1.5), CD8a (53.6.7), Sca1 (Ly6A/E, D7), cKit (CD117, ACK4 or 2B8), CD150 (TC15-12F12.2), CD105 (MJ7/18), CD16/32 (24G2), CD127 (A7R34), CD135 (A2F10), Ly6D (49-H4), CD21/CD35 (7G6), CD23 (B3B4), CD93 (AA4.1), CD24 (M1/69), CD43 (S7), CD45.2 (S450-15-2), CD45.1 (A20), IgM$^b$ (AF6-78), IgD (11-26 c.2a), CD138 (281.2), IgG1 (X56), CD25 (3C7), CD44 (IM7). Secondary staining used streptavidin PE-Texas-Red (Invitrogen). See Supplementary Table 4 for antibody dilutions and catalogue numbers for commercial antibodies. FACS-Gal analysis was performed using warm hypotonic loading of fluorescein di β-D-galactopyranoside (Molecular Probes) on single cells[50] followed by immunophenotyping using relevant surface antigens as defined in Supplementary Table 1. Cells were analysed using a LSR II or FACS Canto flow cytometer (Becton Dickinson) or sorted using a FACSAria II (Becton Dickinson) flow cytometer after antibody staining and lineage selection or depletion using anti-biotin beads and LS columns (Miltenyi Biotec). Data were analysed using FlowJo software (Version 8.8.7, Tree Star).

**Splenic B-cell culture**. Purified and CTV labelled splenic B cells were cultured with either AffiniPure F(ab')₂ Fragment Goat Anti-Mouse IgM μ Chain Specific (20 μg/ml; Jackson Immunoresearch), CD40L (produced in-house)[51] supplemented with IL4 (10 ng/ml; R&D systems) and IL5 (5 ng/ml; R&D systems) to assess T-cell dependent responses, or LPS (25 ug/ml; Difco) to assess T-cell independent responses, and analysed by flow cytometry.

**Analysis of publicly available RNA-seq datasets**. FASTQ files containing RNA-seq profiles of pre–proB cells from $Ebf1^{\Delta/\Delta}$ (GSM2879293, GSM2879294, GSM2879295), pro-B cells from $Pax5^{\Delta/\Delta}$ (GSM2879296, GSM2879297, GSM2879298) and control populations from wild-type mice (GSM2879299, GSM2879300, GSM2879301). Reads were aligned to the mm10 genome using Rsubread's align function and read counts were summarised at the gene level as for the primary samples (See Supplementary Methods)[52]. Genes were filtered from downstream analysis using edgeR's filterByExpr function and library sizes were TMM normalised. Counts were transformed to log2-CPM and the mean-variance relationship estimated using the *voom* function in limma[53]. Heatmaps were generated using heatmap.2 function in gplots. Genes were tested for differential expression using linear modelling in limma 3.38.2[54]. Gene set testing was performed using *camera*[55] and barcode plots were generated with limma.

For single cell RNA-seq analysis raw counts corresponding to single-cell RNA-seq of wild-type mouse CLPs, pre–proB and CD19$^+$ B lymphoid progenitor cells were downloaded from Gene Expression Omnibus repository, accession GSE114793. Raw counts were filtered to remove low expressed genes and cells with low cell quality. Read counts were then L1 normalised such that the sum of expression values for each cell sums to 1. Library sizes were then normalised by median counts per cell. Normalised read counts were then imputed using the MAGIC algorithm[56] (Rmagic v 1.4.0) with the settings $t = 11$, $k = 30$ with other parameters set at default values. The tSNE visualisation of the first 20 principal components of the imputed values was obtained using Rtsne (v 0.15) package with the following parameters: perplexity parameter = 80, momentum of 0.5 for the first 250 iterations and a final momentum of 0.8. The learning rate of the tSNE was set to 200 with an exaggeration factor of 12. PCA initialisation was disabled. All of the analysis was performed in R version 3.6.1.

**ATAC-seq analysis**. ATAC-seq[57] was performed on sorted pre–proB, proB and preB populations. Briefly, $5 \times 10^4$ nuclei were fragmented by sonication for 30 min at 37 °C and the DNA purified prior to amplification with indexing primers (HiFi Ready Mix, Kapa Biosciences) for 13 PCR cycles followed by quality assessment by Bioanalyser. High quality libraries were size selected (150–700 base pairs) and sequenced using a high output paired end 75 base pair kit on the NextSeq 500 (Illumina) to a minimum of 50 million reads. ATAC-seq reads were aligned to mm10 genome using Bowtie2[58] (http://bowtie-bio.sourceforge.net/bowtie2/index.shtml accessed 6th March 2017). Peak calling was performed using MACS2[59]. Intersections of genetic coordinates were performed using Bedtools (http://bedtools.readthedocs.io/en/latest/ accessed 6 March 2017). Heatmaps of unique peaks were generated using pHeatmap in R. These data have been deposited in Gene Expression Omnibus database (accession number GSE132852).

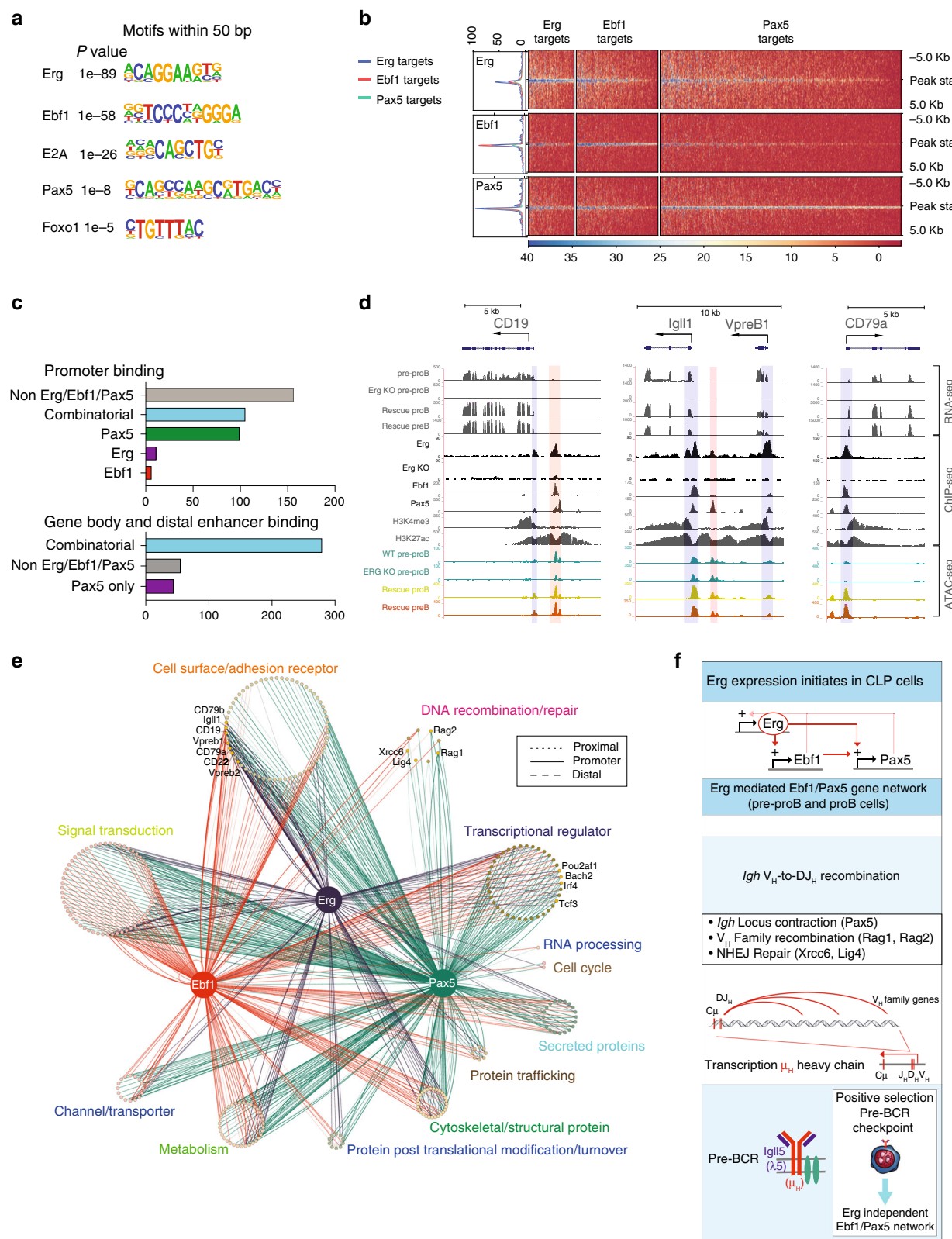

**Visualisation of RNA-seq, ChIP-seq and ATASeq data.** RNA-seq, ChIP-seq and ATAC-seq files were converted to BigWig files using deepTools (version 2)[60] and uploaded to Cyverse (www.cyverse.org) for visualisation in UCSC Genome Browser[61] (genome.ucsc.edu).

**Gene network analysis.** All Ebf1-, Pax5- and Erg-ChIP-seq peaks mapping to differentially expressed genes in $Rag1Cre^{T/+};Erg^{\Delta/\Delta}$ pre–proB cells within 10 kb of the TSS were identified. Peaks inside the gene body were annotated as

"proximal targets", peaks overlapping the TSS were labelled as promoter regulated targets, peaks less than 3 kb upstream or downstream of the TSS were labelled as putative promoter regulated targets, peaks more than 3 kb upstream or downstream TSS were labelled as putative distal targets. Gene Ontogeny (GO) annotation of differentially expressed genes was performed and underwent expert manual curation. The network was constructed using[62] CRAN package, and exported to Cytoscape[63] for customisation using RCy3[64] R/Bioconductor package.

**Fig. 6 The Erg-mediated Ebf1 and Pax5 gene regulatory network. a** Whole genome HOMER motif discovery underlying Erg bound regions in proB cells. **b** Heatmap of Erg, Ebf1 and Pax5 binding to differentially expressed genes in Rag1Cre$^{T/+}$;Erg$^{\Delta/\Delta}$ pre–proB cells centred around the transcriptional start site (TSS) ±5.0 kB (see Supplementary Data 2 for all annotated ChIP binding sites). **c** Erg, Ebf1 and Pax5 binding to annotated regions of genes differentially expressed between Rag1Cre$^{T/+}$;Erg$^{\Delta/\Delta}$ and Erg$^{fl/fl}$ pre–proB cells. **d** RNA-seq gene expression at CD19, Igll1, VpreB1 and CD79a loci, with ChIP-seq of Erg, Ebf1, and Pax5 binding, H3K4me3 promoter mark, H3K27ac promoter and enhancer mark in wild-type proB cells, and ATAC-seq of Erg$^{fl/fl}$ pre–proB cells, Rag1Cre$^{T/+}$;Erg$^{\Delta/\Delta}$ pre–proB (Erg KO pre–proB), and Erg-deficient proB (Rescue proB) and preB (Rescue preB) cells in Rag1Cre$^{T/+}$;Erg$^{\Delta/\Delta}$;Igh$^{VH10tar}$/+mice rescued with a functionally rearranged Igh allele. Erg, Ebf1 and/or Pax5 binding promoters (blue shading). Erg, Ebf1 and/or Pax5 binding to enhancer regions (pink shading). **e** The Erg-dependent Ebf1 and Pax5 transcriptional network in proB cells with binding of each transcription factor shown to annotated promoter, proximal and distal gene regions of differentially expressed genes in Rag1Cre$^{T/+}$;Erg$^{\Delta/\Delta}$ pre–proB cells (full list in Supplementary Data 2). **f** Summary of the Erg-dependent Ebf1 and Pax5 transcriptional network in V$_H$-to-DJ$_H$ recombination and pre-BCR formation.

**Hi-C analysis.** For in situ Hi-C analysis[13,65], primary immune cell libraries were generated in biological duplicates for each genotype. An Illumina NextSeq 500 was used to sequence libraries with 80 bp paired-end reads to produce libraries of sizes between 42 million and 100 million valid read pairs. Each sample was aligned to the mm10 genome using the diffHic package v1.14.0[66], which utilises cutadapt v0.9.5[67] and bowtie2 v2.2.5[58] for alignment. The resultant BAM file was sorted by read name, the FixMateInformation command from the Picard suite v1.117 (https://broadinstitute.github.io/picard/) was applied, duplicate reads were marked and then re-sorted by name. Read pairs were determined to be dangling ends and removed if the pairs of inward-facing reads or outward-facing reads on the same chromosome were separated by less than 1000 bp for inward-facing reads and 6000 bp for outward-facing reads. Read pairs with fragment sizes above 1000 bp were removed. An estimate of an alignment error was obtained by comparing the mapping location of the 3′ segment of each chimeric read with that of the 5′ segment of its mate. A mapping error was determined to be present if the two segments were not inward-facing and separated by less than 1000 bp, and around 1–2% were estimated to have errors. Differential interactions (DIs) between the three different groups were detected using the diffHic package[66]. Read pairs were counted into 100 kbp bin pairs. Bins were discarded if found on sex chromosomes, contained a count of less than 10, contained blacklisted genomic regions as defined by ENCODE for mm10[68] or were within a centromeric or telomeric region. Filtering of bin-pairs was performed using the filterDirect function, where bin pairs were only retained if they had average interaction intensities more than 5-fold higher than the background ligation frequency. The ligation frequency was estimated from the inter-chromosomal bin pairs from a 500 kbp bin-pair count matrix. The counts were normalised between libraries using a loess-based approach. Tests for DIs were performed using the quasi-likelihood (QL) framework[69] of the edgeR package. The design matrix was constructed using a layout that specified the cell group to which each library belonged and the mouse sex. A mean-dependent trend was fitted to the negative binomial dispersions with the estimateDisp function. A generalised linear model (GLM) was fitted to the counts for each bin pair[70], and the QL dispersion was estimated from the GLM deviance with the glmQLFit function. The QL dispersions were then squeezed towards a second mean-dependent trend, using a robust empirical Bayes strategy[71]. A P value was computed against the null hypothesis for each bin pair using the QL F test. P values were adjusted for multiple testing using the Benjamini–Hochberg method. A DI was defined as a bin pair with a false discovery rate (FDR) below 5%. DIs adjacent in the interaction space were aggregated into clusters using the diClusters function to produce clustered DIs. DIs were merged into a cluster if they overlapped in the interaction space, to a maximum cluster size of 1 Mbp. The significance threshold for each bin pair was defined such that the cluster-level FDR was controlled at 5%. Cluster statistics were computed using the csaw package v1.16.0[72]. Overlaps between unclustered bin pairs and genomic intervals were performed using the InteractionSet package[73]. Plaid plots were constructed using the contact matrices and the plotHic function from the Sushi R package[74]. The colour palette was inferno from the viridis package (https://github.com/sjmgarnier/viridis accessed 30 March 2018) and the range of colour intensities in each plot was scaled according to the library size of the sample. The plotBedpe function of the Sushi package was used to plot the unclustered DIs as arcs where the z-score shown on the vertical access was calculated as -log$_{10}$(p-value). These data have been deposited in Gene Expression Omnibus database (accession number GSE133246).

**Fluorescence in situ hybridisation.** Cultured B-cell progenitors were resuspended in hypotonic 0.075 M KCl solution and warmed to 37 °C for 20 min. Cells were pelleted and resuspended in 3:1 (vol/vol) methanol:glacial acetic acid fixative. Fixed cells were dropped onto coated Shandon$^{TM}$ polysine slides (ThermoFisher Scientific) and air dried. The cells were hybridised with FISH probes (Creative Bioarray) at 37 °C for 16 h beneath a coverslip sealed with Fixogum (Marabu) after denaturation at 73 °C for 5 min. Cells were washed at 73 °C in 0.4× SSC/0.3%NP$_{40}$ for 2 min followed by 2× SSC/0.1%NP$_{40}$ for less than 1 min at room temperature and air dried in the dark and cover slipped. Images of nuclei were captured on an inverted Zeiss LSM 880 confocal using a 63×/1.4 NA oil immersion objective. Z-stacks of images were then captured using the lambda scan mode, a 405 and a multi-band pass beam splitter (488/561/633). The following laser lines were used: 405, 488, 561 and 633 nm. Spectral data were captured at 8 nm intervals. In all

cases, images were set up with a pixel size of 70 nm and an interval of 150 nm for z-stacks. Single dye controls using the same configuration were captured and spectra imported for spectral unmixing using the Zen software (Zen 2.3, Zeiss Microscopy). Unmixed data were then deconvolved using the batch express tool in Huygens professional software (Scientific Volume Imaging). Images were analysed using TANGO software[75] after linear deconvolution. Nuclear boundaries were extracted in TANGO using the background nuclear signal in the Aqua channel. A 3D median filter was applied and the 3D image projected with maximum 2D image projection for nuclei detection using the Triangle method for automated thresholding in ImageJ[76]. Binary image holes were filled and a 2D procedure implemented to separate touching nuclei using ImageJ 2D watershed implementation. The 2D boundaries of the detected nuclei were expanded in 3D and inside each 3D delimited region, Triangle thresholding was applied to detect the nuclear boundary in the 3D space. Acquired images from immunofluorescent probes were first filtered using 3D median and 3D tophat filter to enhance spot-like structures followed by application of the "spotSegmenter" TANGO plugin with only the best four spots having the brightest intensity kept for analysis. The spots identified by TANGO were manually verified against the original immunofluorescent image to identify and record the correct distance computed by TANGO between the aqua and 5-Rox immunofluorescent probes for both Igh alleles within a nucleus.

**Statistical analysis.** Student's unpaired two-tailed t tests were used using GraphPad Prism (GraphPad Software) unless otherwise specified. Unless otherwise stated, a P value of <0.05 was considered significant.

Details of reagents and software packages used are provided in Supplementary Table 4.

**Reporting summary.** Further information on research design is available in the Nature Research Reporting Summary linked to this article.

## Data availability
The following datasets analysed in the current study are available at the NCBI Gene Expression Omnibus, accession GSE132852 (ATAC-seq), GSE132853 (ChIP-seq), GSE132854 (RNA-seq), GSE133246 (Hi-C). The source data underlying Figs. 1b, d–f, 2a, b, d, 3a–d, f, g, 4d, 5b, d–f, Supplementary Figs. 2a–c, 3b, c are provided in the Source Data file. The data supporting this study are available in the Article, Supplementary Information, Source Data or available from the authors upon reasonable requests. Source data are provided with this paper.

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

## Acknowledgements

We thank Janelle Lochland, Jason Corbin, Jasmine McManus, Melanie Salzone, Carolina Alvarado, Keti Stoev, Nicole Lynch and Shauna Ross for skilled assistance. We thank Professor Robert Brink for the $V_H$10tar knock-in mouse line. This work was supported by Program Grants (1113577, 1016647, 1054618, 1054925), Project Grant (A.P.N. 1060179, 1122783, M.J.H. 1186575, 1159658), Fellowship (D.M.T. 1060675, S.L.N. 1155342, WSA 1058344, T.M.J. 1124081, M.J.H. 1156095), C.R.B. Blackburn Scholarship (M.S.Y.L., Australian National Health and Medical Research Council jointly with Royal Australasian College of Physicians) and Independent Research Institutes Infrastructure Support Scheme Grant (361646) from the Australian National Health and Medical Research Council, the Australian Cancer Research Fund and Victorian State Government Operational Infrastructure Support. Y.C.C. was supported through Maddie Riewoldt's Vision. M.J.H. was supported through the Leukemia and Lymphoma Society of America (LLS SCOR 7001-13). The MAGEC laboratory was supported by the Australian Phenomics Network and the Australian Government through the National Collaborative Research Infrastructure Strategy Program.

## Author contributions

Conceptualisation, A.P.N., M.S.Y.L, A.J.K., T.M.J., M.A.D., R.S.A., K.R., D.M.T., G.K.S., M.J.D., S.L.N. and W.S.A.; Methodology, A.P.N., M.S.Y.L, C.C.B., O.G., T.M.J., T.B., A.J.K., M.J.H., M.A.D., R.S.A., K.R., D.M.T., G.K.S., M.J.D., S.L.N. and W.S.A.; Investigation, A.P.N., H.D.C., S.H., K.B., T.M.J., M.S.Y.L., C.C.B., O.G., Y.C.C., T.B., L.D., C.D.H., H.I., S.M., E.V., A.J.K., R.F., A.G., T.W., K.R., G.K.S., M.J.D.; Formal analysis, A.P.N., H.D.C., S.H., M.S.Y.L., O.G., C.C.B., Y.C.C. T.B., K.R., M.J.D., S.L.N; Writing—Original Draft, A.P.N.; Writing—Review & Editing, A.P.N., H.D.C., S.H., M.S.Y.L., A.J.K., G.K.S., S.L.N., and W.S.A.; Funding Acquisition, A.P.N. and W.S.A.; Supervision, A.P.N., M.A.D., D.M.T., G.K.S., M.J.D., S.L.N. and W.S.A.

## Competing interests

The authors declare no competing interests.
