## [Peer Review File · Nature Communications]

Reviewers' Comments:

Reviewer #1:

Remarks to the Author:

Ng et al. describe the analysis of B lymphopoiesis in mice in which the transcription factor gene *Erg* is conditionally inactivated by *Rag1*-driven *Cre* expression. The authors observe a developmental block at the pre-pro-B cell stage and a reduced V to DJ recombination of the immunoglobulin heavy chain (*Igh*) locus. The authors use 3D FISH and Hi-C experiments to show that long-range interactions across the *Igh* locus are reduced. The authors find that many B-lineage genes, including *Ebf1* and *Pax5*, are not expressed in *Erg*-deficient pre-pro-B cells. They also find that the complementation with a rearranged *Igh* knock-in allele overcomes the developmental block and allows for normal expression of *Ebf1* and *Pax5*. Finally, the authors show that *Erg* binds to regulatory regions of many B lineage genes, including *Ebf1* and *Pax5*.

This study provides convincing evidence for a role of *Erg* in early B lymphopoiesis. However, some conclusions of the molecular analysis require additional experimental support.

Specific comments:

The authors claim that *Erg* initiates a transcriptional network involving *Ebf1* and *Pax5*. Surprisingly, the expression of a rearranged *Igh* chain is sufficient to overcome the developmental block and allows for a normal expression of *Ebf1* and *Pax5* despite the lack of *Erg*. Are other closely related Ets family members compensating for the lack of *Erg* in *Igh*-rescued mice?

In the ChIP-seq analysis of *Erg* binding at the *Ebf1*, *Pax5* and *Tcf3* loci in pre-pro-B cells shown in Figure 4c, the authors use *Erg* KO thymocytes as a control for the specificity of the *Erg* ChIP. The majority of thymocytes, including CD4CD8 double positive and CD4 and CD8 single positive thymocytes, do not express *Erg* (see Figure 1b). Therefore, the authors should perform an *Erg* ChIP-seq analysis with *Erg* KO pre-pro-B cells. This control would also be more appropriate for the assessment of *Erg* binding at the *Cd19*, *Igll1* and *Cd79a* loci shown in Figure 6b.

In Figures 2 and S3, the authors show that *Erg* can bind to the μ A region of the $iE\mu$ enhancer. The deletion of the μ A region did not result in a perturbation of B cell development. The authors should perform an *Erg* ChIP with μ A-deleted cells to examine whether *Erg* can bind to other regions of the $iE\mu$.

To confirm that *Ebf1* and *Pax5* are transcription factors downstream of *Erg* in pre-pro-B cells, the authors transfected (or transduced?) *Erg*-deficient progenitor cells with *Ebf1* or *Pax5* retroviral expression constructs. In Figure 5d and 5e, the authors show a partial rescue of V to DJ recombination and CD19 expression in *Ebf1*-transfected cells. It is unclear whether they sorted *Ebf1*-expressing cells for the analysis. The authors should also use a rescue with an *Erg* expression construct as a positive control.

In several figure legends and in the text, the authors use the term B-cell progenitors. The authors should clarify which cell type they refer to.

Reviewer #2:

Remarks to the Author:

Ng et al show that the ETS transcription factor *ERG* plays a non-redundant role in early B-cell development by setting up a transcriptional program with two other stage specific TFs, *EBF1* and *PAX5* in downstream VDJ recombination. To do so, they generate and use a KI mouse with a conditional reporter cassette that is also used to disrupt expression in CLPs (*Rag1-Cre*). Overall a convincing case is made both phenotypically and molecularly for *ERG* as a driver of normal B-cell development in conjunction with at least two other stage specific TFs. Showing that a transcription factor is involved preferentially in early and not late stages of B cell differentiation isn't of itself particularly interesting. We know such specific factors must exist, so it's not changing any paradigm, and their claim that *Erg* is the "master" transcription factor can only stand until someone shows there's another one upstream - and someone will sooner or later in my opinion. Toning down the language would be appropriate.

Figures

Figure 1: Expression of *lacZ* in early stem and B/T cell progenitors with an inflection in Pro Bs and accumulation of pre-Pro B cells in the bone marrow of *Rag1-Cre/ERG* KO mice.

Comments-

The Y-axis label in Figure 1b is not intuitive and confusing.

Figure 1d. If you have B cell data, could you add it to the figure?

Figure 1e-clarify labels- RNA-, ChIP-, ATAC- etc.

Figure 1f. Can you put the numbers in the FACS plots?
Figure 1f. Can you change the error bars from SEM to SD?

Figure 2: Absence of VH-to-DJH rearrangement/locus contraction in IgH alleles of B220+ cells in Rag1CreT/+;ErgΔ/Δ mice is shown both by PCR and FISH along with the absence of an association between ERG binding and reduced IgH locus interactions by integrating ChIP and Hi-C data and preserved chromatin accessibility across the IgH locus. An ERG bound E_μ enhancer is shown not to account for the absence of VH-to-DJH recombination.

In Fig 2C Erg ChIP data is shown as a cartoon with reference to Fig S4 for detail, but Fig S4 does not show Erg ChIP for the Igh enhancer regions - it shows Erg ChIP for the upstream Vh genes, so as far as I can see they haven't shown the actual Erg ChIP data for the enhancer regions of IgH. So where does the Erg ChIP cartoon in Fig 2C come from?

A cartoon showing the approximate positions of the primers used to semi-quantify V(D)J-recombination in Fig 2a and elsewhere would be helpful to the reader as a Suppl Fig. or as a panel in Fig 2.

Comment- the preservation of chromatin accessibility at the IgH locus would also indicate that the ERG/EBF1/PAX5 network is not involved in setting up this process in these cells?

Figure 3: IgH^{VH10tar} knock-in allele rescues the B cell phenotype in Rag1-Cre/ERG KO mice.

Figure 3a. Could the authors also use a Rag1CreT/+, IgH^{VH10tar}/+ mouse as a control?

Figure 3d and Figure 3e. The authors are careful to say that Erg is not required for differentiation into PC or for switching. They avoid saying that it has no quantitative impact, but n should be ≥5. Their problem in these experiments is that the proliferation kinetics for Igh-V10tar cells is going to be slightly different than for their other genotypes - this is something demonstrated in ref 32 and can also be seen clearly in the greater responsiveness of the IgH-V10tar cells to LPS in Fig 3d. So, the only way they can quantitatively compare differentiation between genotypes in vitro is to plot differentiation into PC and CSR per cell division cohort - as they do in Fig 3d, but not in 3e- why?

Can the authors add representative FACS plots? This might have to be a supplementary figure.

Figure 3f. The legend uses squares while the graph uses circles, could you change them so that they are the same

Figure 4: Gene expression and ChIP-seq to demonstrate direct regulation and absence of EBF1/PAX5 expression in Rag1-Cre/ERG KO B cell progenitors with co-binding at differentially expressed genes.

Figure 4b. The heatmap is hard to interpret without WT to compare.

Figure 5: Gene expression comparisons using publicly available datasets WT/EBF1/PAX5 show interdependency of expression. V to D recombination in Rag1-Cre/ERG KO B cells was only partially rescued (EBF1) or not rescued (PAX5) by over-expression of EBF1/PAX5 showing the need for co-dependency with ERG at the apex.

Comment- are there single cell expression data the B- cell hierarchy (showing co-expression of these factors, strength of expression and cell trajectories) to complement these data?

Figure 5d. Could the authors change the y-axis label of this graph? The figure legend makes no specific mention of GFP.

Figure 6: Combinatorial binding of ERG/EBF1/PAX5 at differentially expressed genes.

Comment- Do these TFs bind each other's loci- after all each was differentially expressed in the absence of the other- i.e. is this a fully connected network triad or do these factors regulate each other indirectly and then act combinatorially on downstream targets?

Figure 6A. What does 'other' mean? Should you split this up into the respective subsets?

Figure S1. The FACS plots are shifted relative to the labels, could you fix this?

Figure S2. Could the authors change the error bars from SEM to SD?

Figure S3b. Could the authors repeat these experiments? There are not enough numbers in these experiments to justify these conclusions.

Figure legends

Figure 2C. This was a little confusing, if possible, could you make it simpler

Figure 4D. Could the authors specify which population of B cell precursors was used to generate this western blot?

Figure 5D. The legend uses the term transfected, should this be transduced?

Text

Page 3, line 50, remove the word 'the'

Page 8, line 169, the word 'Igh' is in a strange font

Page 9, line 206, could the authors speculate as to why the Vtar allele, while increasing B cell numbers doesn't restore them to wild type levels

Page 13, line 286. The authors describe transfecting cells with a MCSV, should this be transduced?

Page 13, line 298, please delete the word 'the'

Page 13, line 315. Could the authors describe Figure 6C in greater detail in the manuscript?

Page 34, line 658. LPS at 25mg/mL this seems a little high

Page 37, line 720, the word "these" uses a strange font

Tables

Could the authors add the fluorophores and dilutions used to the key resources table?

Reviewers' comments:

Reviewer #1

Specific comments:

The authors claim that Erg initiates a transcriptional network involving Ebf1 and Pax5. Surprisingly, the expression of a rearranged Igh chain is sufficient to overcome the developmental block and allows for a normal expression of Ebf1 and Pax5 despite the lack of Erg. Are other closely related Ets family members compensating for the lack of Erg in Igh-rescued mice?

There are 18 members of the ETS family. There is no rational approach in the context of our study that could allow us to narrow the focus on a reduced number of ETS family members to identify and fully validate those that may or may not compensate for the lack of Erg in *Igh* rescued mice within a limited time-frame. It was therefore not possible to address this question within the period of revision.

We have added the statement to the discussion “The transcriptional regulators of *Ebf1* and *Pax5* expression during these later stages of B-cell development remain to be defined”.

In the ChIP-seq analysis of Erg binding at the Ebf1, Pax5 and Tcf3 loci in pre-pro-B cells shown in Figure 4c, the authors use Erg KO thymocytes as a control for the specificity of the Erg ChIP. The majority of thymocytes, including CD4CD8 double positive and CD4 and CD8 single positive thymocytes, do not express Erg (see Figure 1b). Therefore, the authors should perform an Erg ChIP-seq analysis with Erg KO pre-pro-B cells. This control would also be more appropriate for the assessment of Erg binding at the Cd19, Igl1 and Cd79a loci shown in Figure 6b.

We have performed Erg ChIP-seq analysis on Erg KO pre-proB cells as requested and the data is included in the revised manuscript (Figure S4B). This confirmed the interpretations of the existing ChIP-seq analysis.

In Figures 2 and S3, the authors show that Erg can bind to the μA region of the $iE\mu$ enhancer. The deletion of the μA region did not result in a perturbation of B cell development. The authors should perform an Erg ChIP with μA -deleted cells to examine whether Erg can bind to other regions of the $iE\mu$.

As discussed in the responses to the editor's comments above, this experiment has been done and the results included in the revised manuscript (Figure S3D).

To confirm that Ebf1 and Pax5 are transcription factors downstream of Erg in pre-pro-B cells, the authors transfected (or transduced?) Erg-deficient progenitor cells with Ebf1 or Pax5 retroviral expression constructs. In Figure 5d and 5e, the authors show a partial rescue of V to DJ recombination and CD19 expression in Ebf1-transfected cells. It is unclear whether they authors sorted Ebf1-expressing cells for the analysis. The authors should also use a rescue with an Erg expression construct as a positive control.

Transduced GFP⁺ cells were sorted for analysis. We have repeatedly attempted the Erg rescue experiment and consistently observed that high level of Erg expression from exogenous vectors is toxic, causing death of B-lineage progenitors in vitro. Thus, this experiment cannot be included. This fact does not affect the interpretations of data presented in the manuscript.

In several figure legends and in the text, the authors use the term B-cell progenitors. The authors should clarify which cell type they refer to.

This has been clarified.

Reviewer #2

Specific comments:

..... their claim that Erg is the “master” transcription factor can only stand until someone shows there's another one upstream - and someone will sooner or later in my opinion. Toning down the language would be appropriate.

We have altered the language to accede to this request and no longer refer to Erg as a “master” transcriptional regulator.

Figure 1:

The Y-axis label in Figure 1b is not intuitive and confusing.

This has been changed to clarify.

Figure 1d. If you have B cell data, could you add it to the figure?

B cell data was not available.

Figure 1e-clarify labels- RNA-, ChIP-, ATAC- etc.

Labels have been added to clarify

Figure 1f. Can you put the numbers in the FACS plots?

Yes, these have been added.

Figure 1f. Can you change the error bars from SEM to SD?

Yes, this has been changed.

Figure 2:

In Fig 2C Erg ChIP data is shown as a cartoon with reference to Fig S4 for detail, but Fig S4 does not show Erg ChIP for the Igh enhancer regions - it shows Erg ChIP for the upstream Vh genes, so as far as I can see they haven't shown the actual Erg ChIP data for the enhancer regions of IgH. So where does the Erg ChIP cartoon in Fig 2C come from?

We had provided the data as BED file coordinates in the original Figure 2C and Supplementary Table 5. In the revised manuscript we have provided the data more fully as a Figure as outlined in the response to Reviewer #1 and the editor (new ChIP-seq data Figure S3D, and Figure S3A in the revised manuscript).

A cartoon showing the approximate positions of the primers used to semi-quantify V(D)J-recombination if Fig 2a and elsewhere would be helpful to the reader as a Suppl Fig. or as a panel in Fig 2.

A cartoon demonstrating the positions of primers used to semi-quantify V(D)J recombination is included in the revised manuscript (Figure 2A).

Comment- the preservation of chromatin accessibility at the IgH locus would also indicate that the ERG/EBF1/PAX5 network is not involved in setting up this process in these cells?

We agree and had indeed included a comment about the Erg/Ebf1/Pax5 network and chromatin accessibility in our original manuscript. This remains unchanged.

Figure 3:

Figure 3a. Could the authors also use a Rag1CreT/+, IgGHVH10tar/+ mouse as a control?

This data is included in Figure 3A.

Figure 3d and Figure 3e. The authors are careful to say that Erg is not required for differentiation into PC or for switching. They avoid saying that it has no quantitative impact, but n should be ≥ 5 . Their problem in these experiments is that the proliferation kinetics for Igh-V10tar cells is going to be slightly different than for their other genotypes - this is something demonstrated in ref 32 and can also be seen clearly in the greater responsiveness of the IgH-V10tar cells to LPS in Fig 3d. So, the only way they can quantitatively compare differentiation between genotypes in vitro is to plot differentiation into PC and CSR per cell division cohort - as they do in Fig 3d, but not in 3e- why?

Can the authors add representative FACS plots? This might have to be a supplementary figure.

We did not show the divisions due to space limitation on the figure in the original manuscript. In the revised manuscript we increased the size of the figure and have provided the divisions in Figure 3F with representative FACS plots included (Figure 3E).

We also note that the reviewer may have misinterpreted the data from reference 32, in which the differences cannot be attributable to the IgHVH10tar allele, but rather the different genetic backgrounds of Rag1 deleted or Prkdc mutated SCID mice.

Figure 3f. The legend uses squares while the graph uses circles, could you change them so that they are the same

We have made this change in the revised manuscript.

Figure 4:

Figure 4b. The heatmap is hard to interpret without WT to compare.

We believe the reviewer has misunderstood the heatmap. This heatmap is a representation of Log2 fold expression change between the Erg KO and wild-type for selected genes. We have clarified this in the figure legend.

Figure 5:

Comment- are there single cell expression data the B- cell hierarchy (showing co-expression of these factors, strength of expression and cell trajectories) to complement these data?

We have undertaken scRNA-seq analysis as described in the response to the editor above. This is included in the revised manuscript Figures 4E and S5.

Figure 5d. Could the authors change the y-axis label of this graph? The figure legend makes no specific mention of GFP.

We have altered the Y-axis label to include GFP in the revised manuscript.

Figure 6:

Comment- Do these TFs bind each other's loci- after all each was differentially expressed in the absence of the other- i.e. is this a fully connected network triad or do these factors regulate each other indirectly and then act combinatorially on downstream targets?

While we already provided this data, to reinforce the demonstration of the interconnected network, we have included in the revised manuscript a Figure in which ChIP-seq data for all 3 transcription factors is shown at each of the *Erg*, *Ebf1* and *Pax5* loci (Figure 4C).

Figure 6A. What does 'other' mean? Should you split this up into the respective subsets?

Now Figure 6C in the revised manuscript, "Other" referred to non Erg/ Ebf1/Pax5 binding. This has been made explicit in the new Figure.

Figure S1. The FACS plots are shifted relative to the labels, could you fix this?

This has been corrected in the revised manuscript.

Figure S2. Could the authors change the error bars from SEM to SD?

Yes, this has been changed in the revised manuscript.

Figure S3b. Could the authors repeat these experiments? There are not enough numbers in these experiments to justify these conclusions.

These experiments have been repeated to n=5 and the data included in the revised manuscript.

Figure legends

Figure 2C. This was a little confusing, if possible, could you make it simpler.

Yes, this has been simplified in the revised manuscript.

Figure 4D. Could the authors specify which population of B cell precursors was used to generate this western blot?

Yes, this has been specified in the revised manuscript.

Figure 5D. The legend uses the term transfected, should this be transduced?

We have altered this to transduced in the revised manuscript.

Text

Page 3, line 50, remove the word 'the'

This has been removed in the revised manuscript.

Page 8, line 169, the word 'Igh' is in a strange font

The font has been corrected in the revised manuscript.

Page 9, line 206, could the authors speculate as to why the Vtar allele, while increasing B cell numbers doesn't restore them to wild type levels

We have added further discussion in the revised manuscript (page 9). We speculate that a significant contributing factor is the reduced number of clones could give rise to a functional BCR. As we are providing one rescue allele compared to the numerous clones that are generated from normal V(D)J recombination, the clonal repertoire that gives rise to a functional BCR complex with a VJ light chain recombination is reduced. This is demonstrated in Figure 3.

Page 13, line 286. The authors describe transfecting cells with a MCSV, should this be transduced?
We have altered this to transduced in the revised manuscript.

Page 13, line 298, please delete the word 'the'
This has been removed in the revised manuscript.

Page 13, line 315. Could the authors describe Figure 6C in greater detail in the manuscript?
Additional detail has been provided in the revised manuscript to describe this diagram, now Figure 6E.

Page 34, line 658. LPS at 25mg/mL this seems a little high
We have confirmed this dose to be 25ug/mL and corrected the manuscript.

Page 37, line 720, the word "these" uses a strange font
The font has been corrected in the revised manuscript.

Tables

Could the authors add the fluorophores and dilutions used to the key resources table?
Yes, this has been included in the revised manuscript as requested.

Reviewers' Comments:

Reviewer #1:

Remarks to the Author:

The revisions addressed my comments in an adequate manner by including new experiments and modifying the text. The revised manuscript is appropriate for publication in Nature Communications.

Reviewer #2:

Remarks to the Author:

The authors have adequately addressed my concerns.